# Chloride intracellular channel (CLIC) proteins function as fusogens

Bar Manori [1,8], Alisa Vaknin[2,8], Pavla Vaňková[3], Anat Nitzan[4], Ronen Zaidel-Bar [4], Petr Man [5], Moshe Giladi [1,6] ✉ & Yoni Haitin [1,7] ✉

Chloride Intracellular Channel (CLIC) family members uniquely transition between soluble and membrane-associated conformations. Despite decades of extensive functional and structural studies, CLICs' function as ion channels remains debated, rendering our understanding of their physiological role incomplete. Here, we expose the function of CLIC5 as a fusogen. We demonstrate that purified CLIC5 directly interacts with the membrane and induces fusion, as reflected by increased liposomal diameter and lipid and content mixing between liposomes. Moreover, we show that this activity is facilitated by acidic pH, a known trigger for CLICs' transition to a membrane-associated conformation, and that increased exposure of the hydrophobic inter-domain interface is crucial for this process. Finally, mutation of a conserved hydrophobic interfacial residue diminishes the fusogenic activity of CLIC5 in vitro and impairs excretory canal extension in *C. elegans* in vivo. Together, our results unravel the long-sought physiological role of these enigmatic proteins.

The Chloride Intracellular Channel (CLIC) family is highly conserved among metazoa, where most organisms express several isoforms/splice variants in a tissue-specific manner[1]. CLICs were discovered as part of a screen to identify novel anion channels[2] and were shown to produce a distinctive chloride conductance during subsequent in vitro reconstitution studies[3,4]. However, CLICs do not fit the paradigm set by classical ion channel proteins, as these proteins exhibit both soluble and membrane-associated forms[5]. Indeed, sequence analysis revealed a weak yet significant homology to the Ω-class of glutathione-S-transferases (GSTs)[6], and a perplexing lack of sequence similarity to any other chloride channel families. Moreover, subsequent structural investigations using X-ray crystallography revealed a globular conformation of the soluble form, composed of an N-terminal thioredoxin domain (TRX) followed by a C-terminal α domain[7–13].

Despite the extensive efforts invested in analyzing CLICs' biochemical properties in vitro, our understanding of their physiological role is incomplete. Indeed, unequivocal evidence for the ability of

CLICs to function as chloride channels is still missing[5]. Yet, their participation in a wide range of cellular and physiological processes is well supported. For example, CLICs were shown to contribute to normal kidney function[14], cell division[15], bone resorption[16], angiogenesis[17,18], and hair cells stereocilia function[19].

Intriguingly, mounting reports indicate the possible participation of CLIC members in processes involving membrane remodeling, such as tubulogenesis[17,20], actin-dependent dynamic membrane shaping[21–23], vacuole formation and fusion[20], and endosomal trafficking[14,24]. Notably, a spontaneous mutation of the mouse CLIC5 ortholog results in congenital progressive hearing impairment, vestibular dysfunction, and progressive loss of stereocilia due to aberrant and miscoordinated membrane remodeling[19]. In addition, knockout studies of EXC-4, a *C. elegans* homolog, and mouse CLIC4, demonstrated defects in excretory canal formation[20] and altered angiogenesis[17], respectively. These genotype-phenotype relations, together with the cellular distribution and in vitro ability of CLICs to directly interact

[1]Department of Physiology and Pharmacology, Faculty of Medicine, Tel-Aviv University, Tel-Aviv 6997801, Israel. [2]School of Chemistry, Raymond & Beverly Sackler Faculty of Exact Sciences, Tel Aviv University, 6997801 Tel Aviv, Israel. [3]Institute of Biotechnology of the Czech Academy of Sciences, Division BioCeV, Prumyslova 595, 252 50 Vestec, Czech Republic. [4]Department of Cell and Developmental Biology, Faculty of Medicine, Tel-Aviv University, Tel-Aviv 6997801, Israel. [5]Institute of Microbiology of the Czech Academy of Sciences, Division BioCeV, Prumyslova 595, 252 50 Vestec, Czech Republic. [6]Tel Aviv Sourasky Medical Center, Tel Aviv 6423906, Israel. [7]Sagol School of Neuroscience, Tel Aviv University, Tel Aviv 6997801, Israel. [8]These authors contributed equally: Bar Manori, Alisa Vaknin. ✉e-mail: moshegil@post.tau.ac.il; yhaitin@tauex.tau.ac.il

with membranes[25], raise the possibility that they actively participate in membrane remodeling processes[5]. Along this line, a common feature shared between CLICs is their ability to undergo rapid membrane translocation upon agonist stimulation. Specifically, microglial CLIC1 was shown to exhibit cytosol to plasma membrane translocation following exposure to amyloid-β peptides[26,27]; G protein-coupled receptor-induced plasma membrane translocation was observed in the case of CLIC2 and CLIC4[28,29]; and, intra-nuclear translocation of CLIC4 was observed following exposure of cells to various stimuli[30–32].

Multiple biochemical and biophysical observations suggest that CLICs' transition between the soluble and membrane-bound conformations involves major structural rearrangements[33]. Our recent studies suggest that CLIC family members may undergo a shift in their conformational ensemble in the presence of membranes and oxidative conditions. This shift skews the conformational distribution towards an elongated and hydrophobic form, facilitating oligomerization and possibly membrane translocation[13]. Here, we sought to utilize a reconstituted system to explore the molecular outcome of CLICs' interaction with membranes. Strikingly, we observed a previously undescribed ability of CLICs to facilitate membrane fusion. To further explore this activity, we utilized an array of complementary biochemical, biophysical, and structural approaches, using human CLIC5 as a case study. We demonstrate that CLIC5 directly interacts with membranes and facilitates their fusion. Using mutagenesis, we establish the role of the inter-domain interface as a node crucial for this biological function. Moreover, X-ray crystallography and mass spectrometry analyses show that the inherent flexibility of CLIC5 is a prerequisite for conformational transitions necessary for membrane interaction and fusion. Finally, we demonstrate that an inter-domain interface mutant lacks fusion activity in vitro and results in excretory canal defects in *C. elegans*. Taken together, our results illuminate the physiological role of CLIC5 as a fusogen and the molecular basis underlying these biological processes.

## Results

### CLIC5 directly interacts with liposomes

The exposure of CLICs to membrane mimetics was previously shown to result in their conformational alteration and oligomerization[11,13,34,35]. However, the possible reciprocal effects conferred by CLICs on the membrane were not investigated. Therefore, we established a reconstituted proteoliposomal system. Peculiarly, during these attempts, we identified enhanced turbidity in the CLIC5-containing liposome sample (Fig. 1a), suggesting that CLIC5 directly interacts with the lipids. To further examine this possible interaction, we performed a liposome co-floatation assay, which allows the separation between soluble and liposome-bound proteins[36]. In this assay, the liposomes-protein mixture is subjected to sucrose gradient fractionation by ultracentrifugation. We included the fluorescent octadecyl-rhodamine B chloride (R18) in the liposomal lipid mixture to detect the liposome-containing fraction. Following fractionation, each fraction was subjected to fluorescence measurement and SDS-PAGE analysis (Fig. 1b). Double C2-like domain-containing protein beta (DOC2B), which interacts with phospholipids in a calcium-dependent manner[37], and 14-3-3, an intracellular adaptor protein[38], were used as positive and negative controls, respectively (Fig. 1c). This assay revealed the presence of CLIC5 in the liposome-containing fraction (Fig. 1b), similar to calcium-bound DOC2B (Fig. 1c). In contrast, DOC2B without calcium and 14-3-3 were only detected in soluble fractions, as expected.

In addition, we measured FRET between CLIC5 tryptophan residues, serving as energy donors, and dansyl-PE (phosphatidylethanolamine)-containing liposomes as the FRET acceptors. A similar FRET-based approach was previously used to establish the intimate interaction of CLIC1 with a lipid bilayer[25]. In line with the liposome co-floatation assay, we observed a time-evolving FRET signal between CLIC5 and the labeled liposomes (Fig. 1d, e), similar to the FRET levels

measured using calcium-bound DOC2B (Fig. 1f, g). Together, these data demonstrate that CLIC5 exhibits direct interaction with liposomes.

### CLIC5 facilitates liposomal fusion

The enhanced turbidity of the liposome sample following incubation with CLIC5 may indicate an increase in their size[39]. Therefore, we next sought to characterize any change in particle size using dynamic light scattering (DLS) analysis (Fig. 2). The incubation of liposomes with CLIC5 resulted in a striking increase in particle diameter (Fig. 2a). Incubation of the liposomes with DOC2B demonstrated a similar increase in diameter in a calcium-dependent manner (Fig. 2b, c), while purified 14-3-3 had no appreciable effect (Fig. 2d). The increased diameter measured using DLS indicates that the liposomes may undergo aggregation or fusion in response to incubation with CLIC5[40]. To discern between these possibilities, we resorted to using the R18-based lipid mixing assay[41]. This membrane-resident dye exhibits self-quenching, which is concentration-dependent. If lipid mixing occurs upon the addition of unlabeled liposomes, R18 undergoes unquenching, detected as an increase of its fluorescent signal.

Incubation of CLIC5 with a liposome mixture consisting of a four-fold excess of unlabeled liposomes resulted in a time-dependent R18 unquenching, indicating the occurrence of lipid mixing between liposomes (Fig. 3a). Moreover, R18 unquenching showed dose dependence in response to increasing concentrations of CLIC5, further supporting its causative relations with lipid mixing. As the membrane interaction of CLIC family members was shown to be facilitated by low pH[42], we next examined the pH dependence of the observed lipid mixing events. Importantly, R18 unquenching was substantially enhanced with decreasing pH levels, highlighting the ability of acidic pH to enhance CLIC5-mediated liposome fusion (Fig. 3b). As expected, DOC2B facilitated R18 unquenching in a calcium-dependent manner (Fig. 3c). In contrast, incubation with 14-3-3 did not result in a fluorescence increase, regardless of protein concentration (Supplementary Fig. 1).

Finally, we used a well-established fluorescence-based assay to exclude that R18 unquenching reflects a possible scramblase activity of CLIC5 (Fig. 3d)[43]. In this assay, liposomes containing small (1%) amounts of nitrobenzoxadiazole (NBD)-labeled lipids are incubated with the membrane-impermeable reducing agent dithionite, which reduces solvent-accessible NBD and quenches its fluorescence. In the absence of scramblase activity, only outer-leaflet NBD-labeled lipids are accessible for dithionite reduction, thereby resulting in up to 50% fluorescence quenching. However, scramblase activity, which facilitates the exchange of lipids between leaflets, results in a higher level of fluorescence quenching over time. Following incubation of NBD-containing liposomes with different concentrations of CLIC5, only 50% NBD fluorescence quenching was observed, indicating that CLIC5 does not possess scramblase activity (Fig. 3d). Full quenching was achieved following the addition of Triton X-100, which dissolves the liposome and exposes inner leaflet lipids to the bulk.

The R18 assay was originally developed to investigate membrane fusion events involved in viral entry[41]. However, hemifusion (fusion between only the outer leaflets of two distinct lipid-bilayer membranes) can also result in R18 unquenching and increased liposomal diameter[37]. Indeed, it was previously shown that DOC2B facilitates stalk formation between liposomes without opening a fusion pore that allows content mixing[37]. Therefore, we tested the ability of CLIC5 to induce full-fusion events using a content mixing assay based on carboxyfluorescein fluorescence unquenching[44]. Here, liposomes encapsulating concentrated carboxyfluorescein are mixed with dye-free liposomes. In the event of a complete fusion between the two liposome populations, content mixing results in carboxyfluorescein dilution and fluorescence unquenching. In line with the results

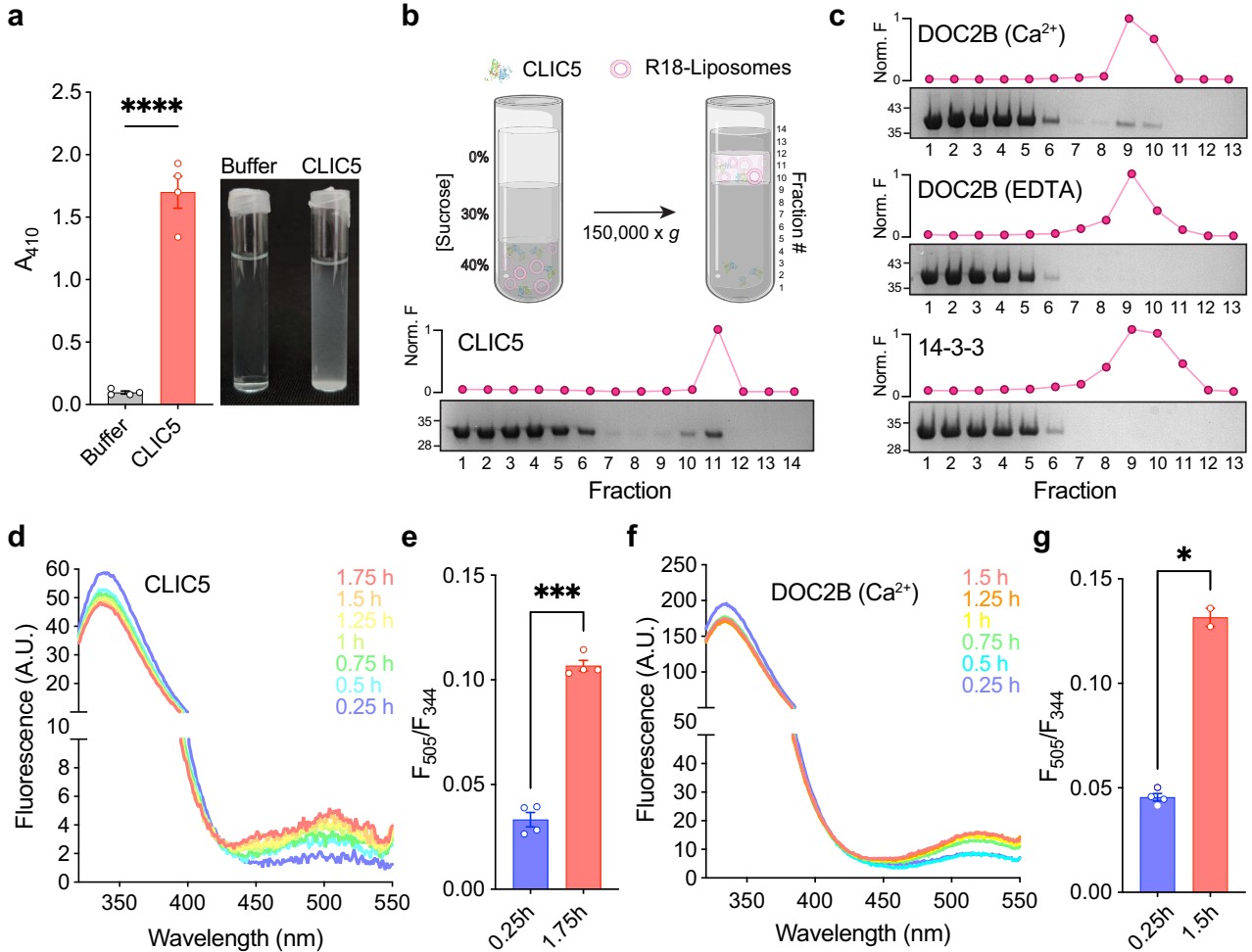

**Fig. 1 | CLIC5 directly interacts with membranes. a** Sample turbidity of liposome samples was measured as 410 nm absorbance ($A_{410}$) in the presence or absence of CLIC5. A representative image of the samples is provided (right). **b** Schematic representation of the liposome co-floatation assay (upper panel; Created with BioRender.com). Representative SDS-PAGE analysis and R18 fluorescence measurement are provided for each fraction, indicating the presence of CLIC5 in direct interaction with the liposomes. Molecular weight markers in kDa are indicated. **c** Control liposome co-floatation assay experiments. DOC2B interacts with the liposomes in a calcium-dependent manner, while 14-3-3 shows no liposomal association. Molecular weight markers in kDa are indicated. **d, f** Averaged fluorescence emission spectra following excitation at $F_{280}$ at the indicated incubation times for CLIC5 (**d**) and calcium-bound DOC2B (**f**). The peak at $F_{344}$ corresponds to tryptophan emission, while the peak at $F_{505}$ represents dansyl-PE emission due to the occurrence of FRET. **e, g** FRET ratio ($F_{505}/F_{344}$) at the indicated time-points for CLIC5 (**e**) and calcium-bound DOC2B (**g**). For all experiments, data are presented as mean ± SEM, $n = 3$–4 independent experiments. Two-sided student's t test was performed for data analysis, ****$P < 0.0001$, ***$P = 0.002$, *$P = 0.0137$.

obtained using the R18 assay, CLIC5 resulted in carboxyfluorescein unquenching in a concentration-dependent manner (Fig. 4a). Conversely, in line with the reported ability of DOC2B to induce stalk formation between liposomes without an open fusion pore[37], it induced only a marginal change in fluorescence that was not calcium-dependent (Fig. 4b). Finally, to eliminate the possibility of fluorescence enhancement caused by liposome leakage, we prepared liposomes containing impermeable NBD-glucose and thoroughly dialyzed the NBD-glucose-loaded liposomes against a buffer devoid of NBD-glucose[43]. Next, the liposomes were incubated with the reducing agent dithionite in the presence of increasing concentrations of CLIC5 (Fig. 4c). Notably, CLIC5 did not cause significant NBD-glucose fluorescence quenching, indicating that the liposomes were intact and impermeable to dithionite. Collectively, our results support the previously unreported functionality of CLIC5 as a fusion-facilitating protein. Importantly, in line with the evolutionary and structural conservation of CLIC family members, this functionality is shared between multiple CLIC paralogs (CLIC2, CLIC6, EXC-4; Supplementary Fig. 2).

## The conserved F34 is involved in CLIC5-mediated membrane fusion

CLICs were previously shown to undergo translocation to different cellular membranes following various stimuli[5]. In the case of CLIC4, the F37D mutant, positioned at the inter-domain interface, failed to translocate to the plasma membrane upon stimulation of $G_{13}$-coupled, RhoA-activating receptors[29]. Interestingly, this residue is strategically positioned at the interface between the TRX- and α-domain. Previously, we showed that the opening of this interface exposes a cryptic hydrophobic surface[13]. Thus, we hypothesized that the membrane translocation processes of CLICs might involve interactions between the exposed interface and the membrane. To examine that, we analyzed the effect of the corresponding mutation, F34D, in CLIC5.

In line with the inability of CLIC4-F37D to undergo membrane translocation, CLIC5-F34D failed to interact with liposomes, as determined using the liposome co-floatation (Fig. 5a) and FRET (Fig. 5b, c) assays. Importantly, as expected, this mutant prevented the large CLIC5-mediated change in liposome diameter measured using DLS (Fig. 5d), as well as lipid- (Fig. 5e) and liposome content mixing (Fig. 5f).

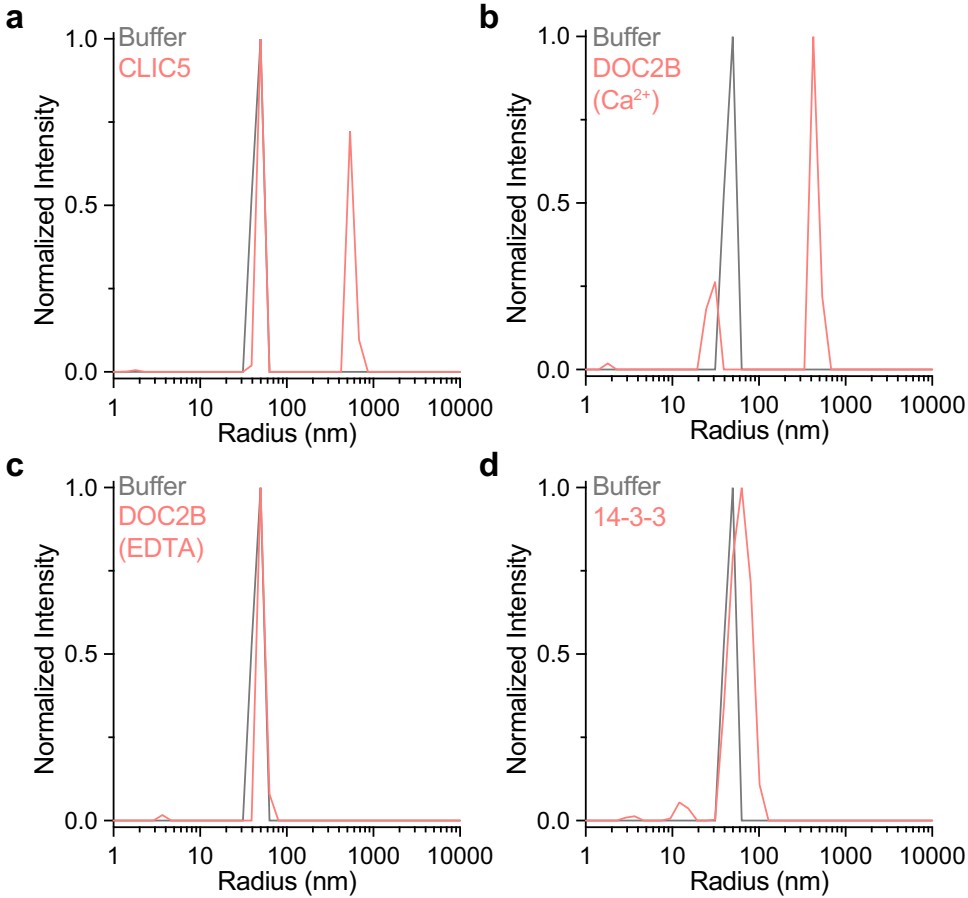

**Fig. 2 | CLIC5 increases the hydrodynamic radius of liposomes. a** Liposomes were incubated with 25 μM CLIC5, as described in the materials and methods section. Following 45 min of incubation, DLS analysis revealed that a population with a significantly increased radius emerged only in the presence of CLIC5. **b–d** Control experiments with calcium-bound DOC2B (**b**), apo DOC2B (**c**), and 14-3-3 (**d**). Only calcium-bound DOC2B resulted in an effect reminiscent of that of CLIC5, as expected.

Together, these data indicate that F34 is critical for the function of CLIC5 as a fusogen.

## Structural analysis of CLIC5 inter-domain interface role in membrane fusion

To elucidate the role of inter-domain interface exposure in CLIC-mediated membrane fusion, we first determined the crystal structure of the F34D mutant (Fig. 6a). Of note, to facilitate the crystallization of F34D, a small flexible loop (residues 57-68) was removed (CLIC5−Δloop). Importantly, a comparison of the CLIC5-WT with or without this loop revealed no functional (Supplementary Fig. 3) or structural perturbation (Supplementary Fig. 4b). Similar to other CLIC proteins, CLIC5-Δloop-F34D consists of tightly packed TRX and α domains, connected by a flexible linker (Supplementary Fig. 4a)[11,13]. Despite the marked functional effect conferred by the F34D mutation, the static high-resolution structure of the compact conformation reveals no significant differences between WT and F34D.

In contrast to the crystalline environment, CLICs exhibit structural flexibility in solution, with an elongated conformation in which the inter-domain interface is exposed[11,13]. Moreover, we showed that the transition to the elongated forms promotes CLIC5 oligomerization, which is further facilitated in the presence of membranes[13]. Therefore, to examine the effect of F34D on protein dynamics in solution and to resolve possible differences in the exposure level of the inter-domain interface, we resorted to hydrogen/deuterium exchange-mass spectrometry (HDX-MS) analysis of CLIC5. In HDX-MS, the exchange of backbone amide hydrogens with deuterium is measured. This process

depends on each residue's involvement in secondary structure formation and solvent accessibility[45]. Following exchange reaction quenching at different time points, the sample undergoes proteolytic digestion, and the deuterium incorporation level is quantitated for each peptide using MS, providing insights into protein dynamics.

First, to assess the structural effects of acidic pH on protein dynamics, leading to enhanced membrane fusion (Fig. 3b), we compared the HDX profile of CLIC5-WT at pH 7.5 and 5.5[46] (Fig. 6b, c and Supplementary Figs. 5, 6). Under acidic conditions, the TRX domain, as well as the interfacial region of the α domain, exhibit markedly increased deuterium uptake. This increase likely reflects increased exposure to the bulk. In accordance, thermal-shift assay (TSA) analysis[47] revealed a pH-dependent increase in the basal fluorescence signal (from $0.01 \pm 0.01$, at pH 7.5, to $0.61 \pm 0.03$, at pH 5.5; $P < 0.001$, $n = 6$), reflecting the binding of the SYPRO Orange dye to hydrophobic protein surfaces prior to the subsequent thermal denaturation (Supplementary Fig. 7). In sharp contrast, the CLIC5-F34D mutant was recalcitrant to an acidic environment, exhibiting diminished alteration in deuterium uptake levels. Notably, while CLIC5-WT and F34D share a similar HDX profile at pH 7.5, the WT protein exhibits a much higher exchange at pH 5.5, indicating that the F34D mutation decreases the conformational heterogeneity of CLIC5 in solution (Fig. 6d, e and Supplementary Fig. 5).

Close inspection of the CLIC5-F34D crystal structure reveals two positively charged residues, which can potentially form a salt-bridge with D34 (Fig. 6a). R37 is within a salt-bridge distance (3.7 Å), while K191 is not captured interacting with D34 in the static structure

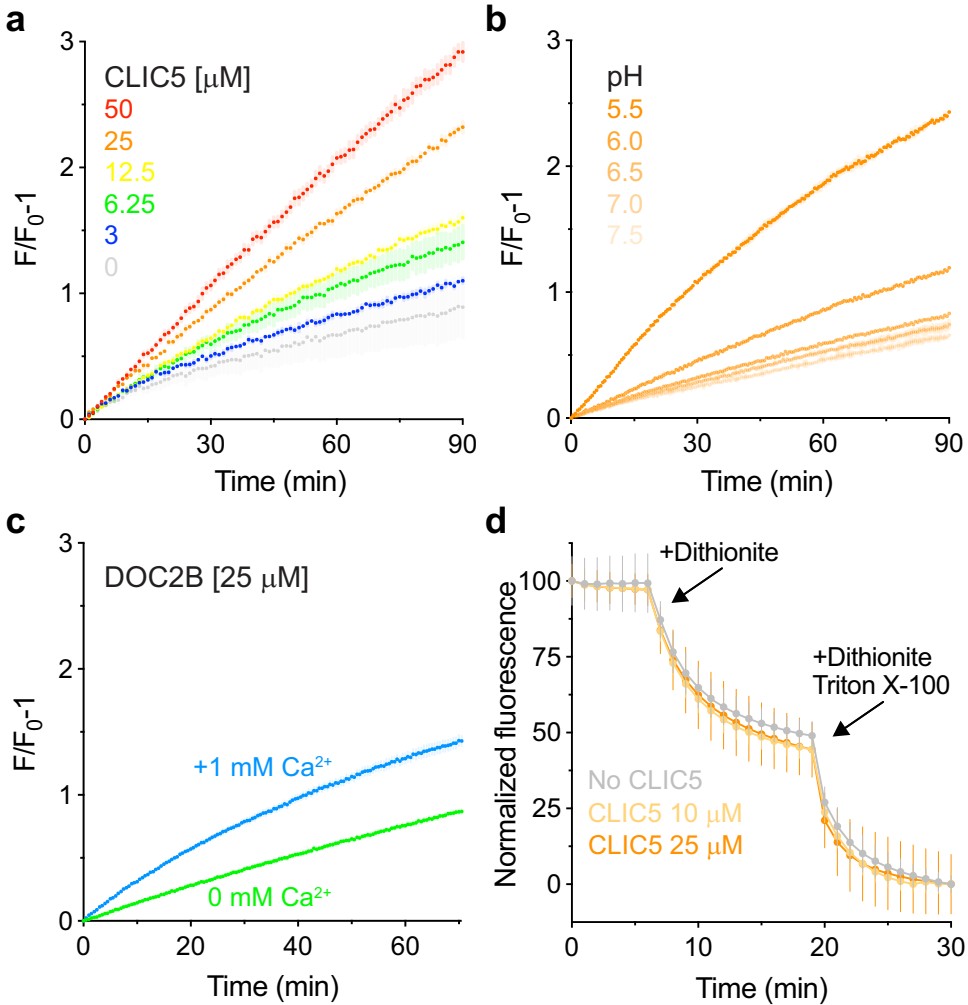

**Fig. 3 | CLIC5 facilitates mixing between liposomal membranes. a** Dose-response analysis of liposomal membrane mixing by CLIC5 using the R18 fluorescence unquenching assay. The time-dependent fluorescence increase reflects membrane mixing between labeled and unlabeled liposomes. **b** pH-dependence of CLIC5-mediated lipid mixing. R18 unquenching was measured in the presence of 25 µM CLIC5 following exposure to different pH values as indicated. **c** R18 unquenching following calcium-dependent lipid mixing by DOC2B, serving as a positive control. **d** Dithionite scrambling assay. CLIC5 does not scramble NBD-PE labeled liposomes, as reflected by the plateau at 50% prior to detergent solubilization with Triton X-100. For all experiments, data are presented as mean ± SEM, $n = 2$–3 independent experiments.

(Supplementary Fig. 8a). Given the dynamic nature of the interface, we performed 1 µsec molecular dynamics simulations of the CLIC5-F34D mutant at both pH 7.4 and pH 5.5 (Supplementary Fig. 8). All the simulations performed here have reached convergence, as reflected by the plateau in root mean square deviation (RMSD) values (Supplementary Fig. 8b). In addition, the per-residue root mean square fluctuation (RMSF) values (representing its spatial fluctuation relative to its average position along the simulation) are similar at both pH values (Supplementary Fig. 8c). These simulations reveal that a salt-bridge interaction forms between D34 and K191 over a substantial fraction of the simulation trajectories (Supplementary Fig. 8d). Of note, this interaction is more stable at pH 7.4. In agreement with the HDX-MS analysis, the TSA analysis of this mutant showed very weak pH-dependent differences in the basal fluorescence signal (from $0.01 \pm 0.01$, at pH 7.5, to $0.06 \pm 0.01$, at pH 5.5; $P < 0.05$, $n = 6$), consistent with low exposure of cryptic hydrophobic surfaces (Supplementary Fig. 7). However, CLIC5-F34D undergoes thermal denaturation at a lower temperature range when exposed to pH 5.5, consistent with the reduced stability of interfacial salt-bridges. Together, our results support the role of inter-domain interface exposure in promoting CLIC-mediated membrane fusion.

## In vivo analysis of the *C. elegans* CLIC ortholog

In the nematode *C. elegans*, the H-shaped excretory canal cell is one of three unicellular tubes that form its excretory system. This cell extends two branched canals along the length of the body from the nose to the tail of the nematode. These canals have an inner lumen closed at its four extremities and collect fluids and waste from the entire body, which then empties into the excretory duct. The EXC-4 protein in *C. elegans* is orthologous to the CLIC protein family and has a crucial role in excretory canal tubulogenesis[20,48,49]. Previous studies showed that excretory canals in *exc-4* null worms do not extend much beyond the pharynx and display large cyst formation. Intriguingly, EXC-4 specifically localizes to distinct membrane domains that undergo substantial remodeling as part of tubular membrane structure formation[20,48].

Notably, the F34 position is conserved between human and *C. elegans* CLICs (F38; Fig. 7a, b, and Supplementary Fig. 9). Therefore, to further establish the involvement of CLICs in membrane remodeling in vivo, we generated an *exc-4* F38D mutant strain via CRISPR and crossed it with an excretory canal reporter strain (Fig. 7c). We assigned an outgrowth score of '1-5' to each canal according to morphological landmarks of the worms (Fig. 7c, control)[49]. A score of '1' indicated no outgrowth, '3' indicated outgrowth to the vulva and '5' indicated

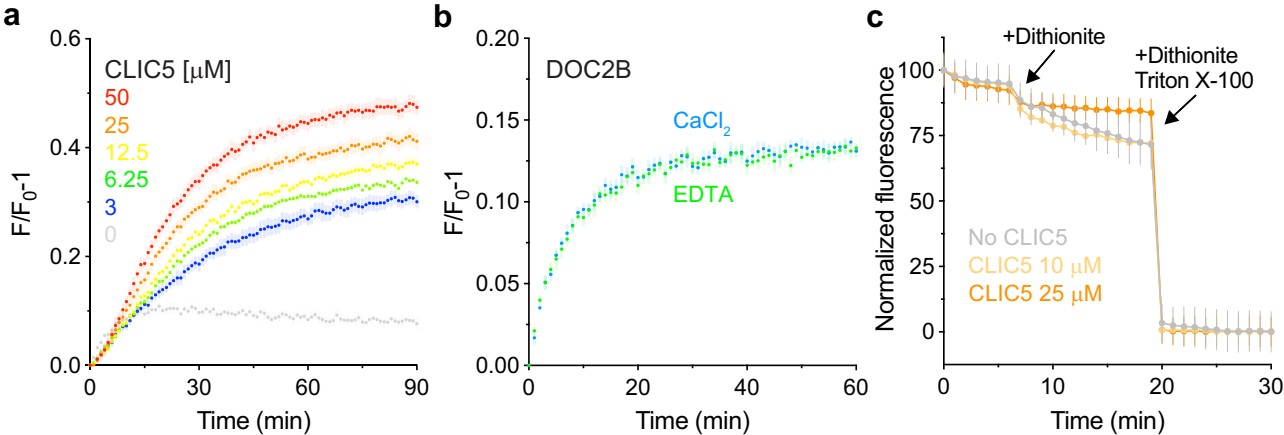

**Fig. 4 | CLIC5 induces liposomal content mixing. a** Dose-response analysis of liposomal content mixing by CLIC5 using the carboxyfluorescein fluorescence unquenching assay. The time-dependent fluorescence increase reflects the dilution of carboxyfluorescein due to the fusion of loaded and unloaded liposomes. **b** DOC2B fails to induce liposomal content mixing in the carboxyfluorescein unquenching assay, regardless of the presence of calcium, consistent with its inability to form a fusion pore. **c** Liposomal content leakage analysis using the fluorescence monitoring of encapsulated NBD-glucose. No leakage from the liposome lumen following incubation with CLIC5 could be detected, indicating that the increase in carboxyfluorescein fluorescence (**a**) does not result from content leakage. For all experiments, data are presented as mean ± SEM, $n = 3–4$ independent experiments.

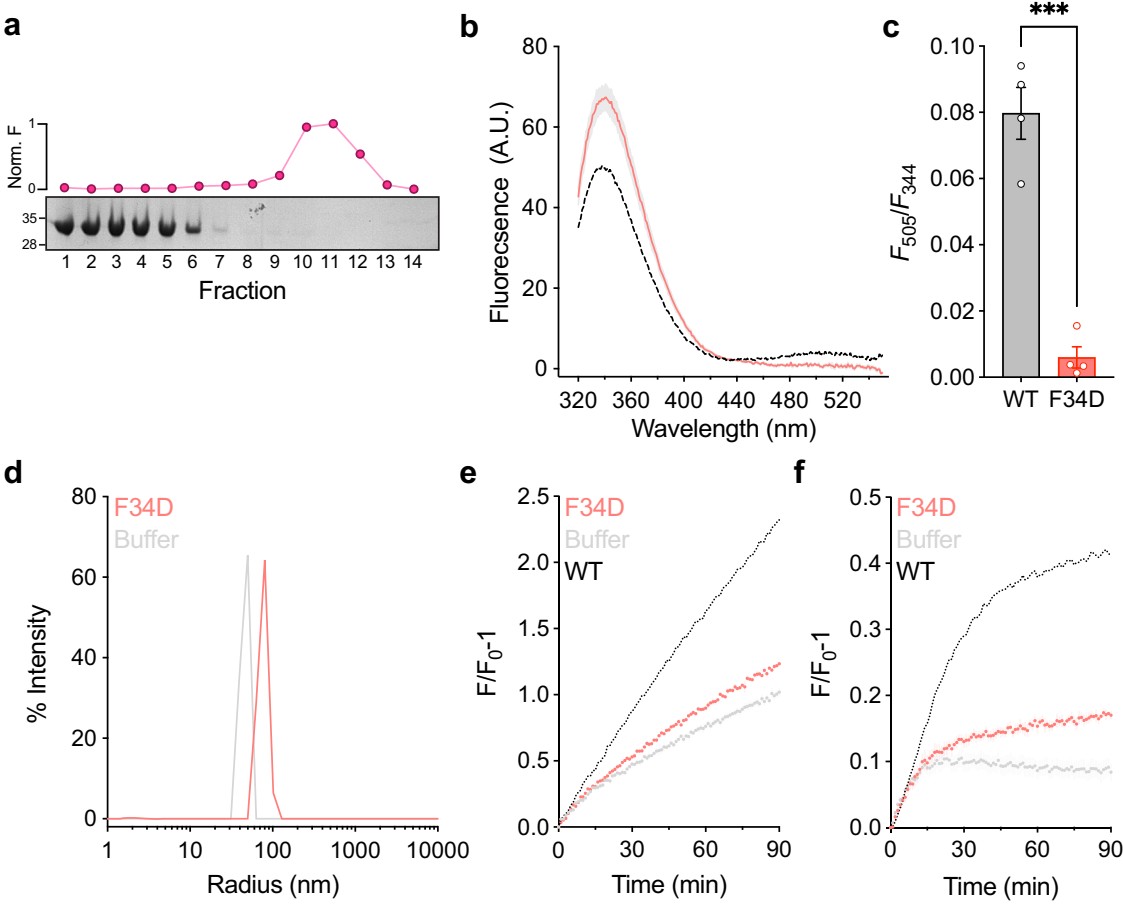

**Fig. 5 | CLIC5-F34D exhibits impaired membrane fusion capability. a** CLIC5-F34D shows reduced interaction with the membrane, as assessed by the co-floatation assay. Representative SDS-PAGE analysis and R18 fluorescence measurement are provided. Molecular weight markers in kDa are indicated. **b** Averaged fluorescence emission spectra following excitation at $F_{280}$ for CLIC5-F34D (red) and CLIC5-WT (black; as in Fig. 1d) after incubation of 75 min. The peak at $F_{344}$ corresponds to tryptophan emission, while the peak at $F_{505}$ represents dansyl-PE emission due to the occurrence of FRET. **c** FRET ratio ($F_{505}/F_{344}$) comparison between CLIC5-WT and CLIC5-F34D, reflecting reduced membrane interaction of CLIC5-F34D. **d–f** CLIC5-F34D exhibits a markedly reduced effect on liposomal diameter as assessed by DLS (**d**), membranal lipid mixing monitored by R18 fluorescence unquenching (**e**), and carboxyfluorescein-mediated liposomal content mixing (**f**). For all experiments, data are presented as mean ± SEM, $n = 3–4$ independent experiments. Two-sided student's t-test was performed for data analysis, ***$P = 0.001$.

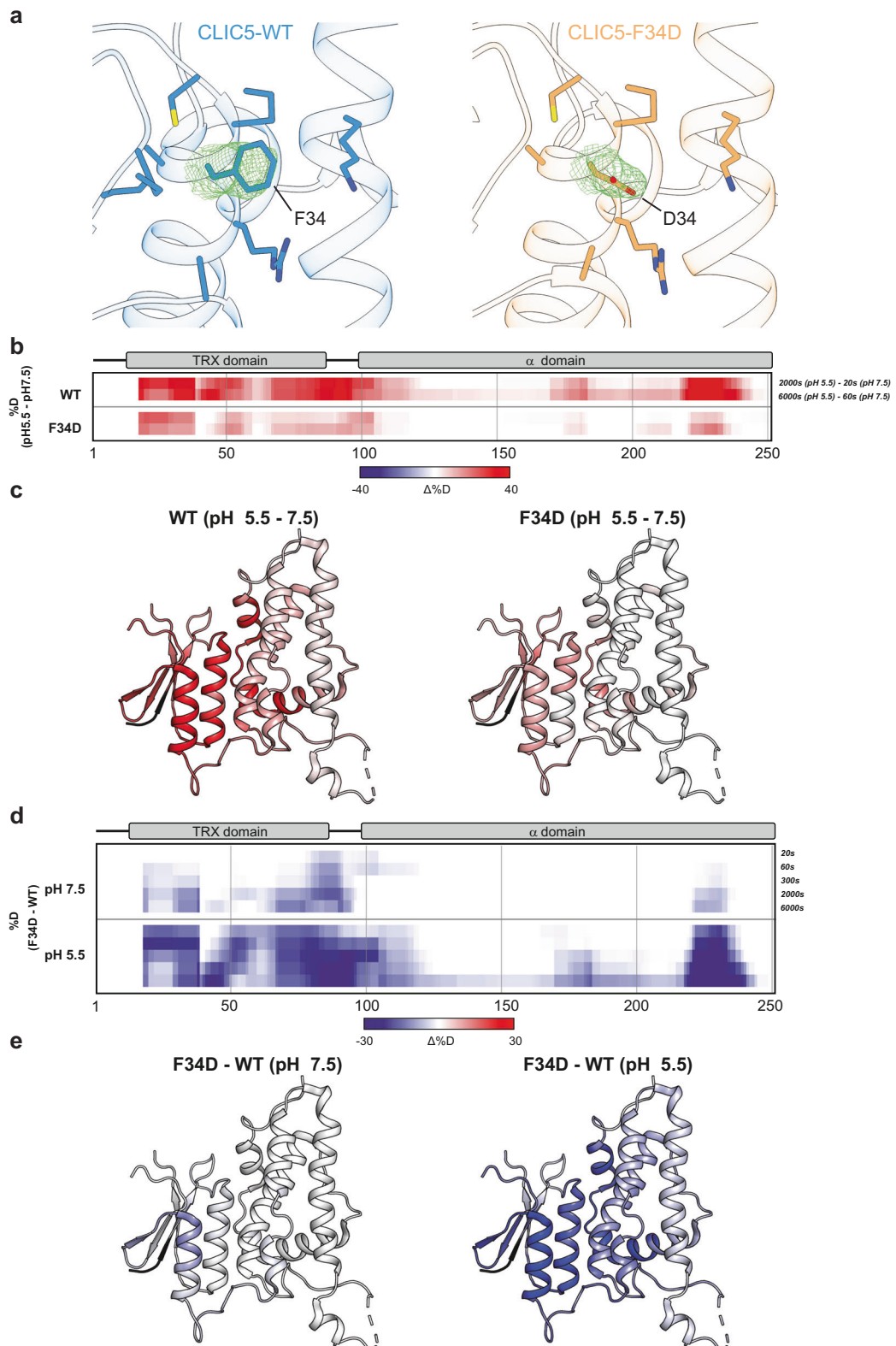

**Fig. 6 | CLIC5 undergoes structural rearrangements upon acidification.**
**a** Crystal structures of CLIC5-Δloop-WT (left) and CLIC5-Δloop-F34D (right). $2F_o$-$F_c$ electron density maps, contoured at 1σ, are provided for position 34, and residues within 5 Å are shown as sticks. **b** The difference in deuteration levels at the indicated time points between pH 5.5 and 7.5 for CLIC5-WT (upper panel) and CLIC5-F34D (lower panel). Due to the difference in the exchange rate at different pH values (exchange 100x slower at pH 5.5 than at 7.5), time points of equal exchange (20 s or 60 s at pH 7.5 and 2000s or 6000 s at pH 5.5, respectively) must be compared. **c** The HDX difference between the pH levels tested at the final time point is mapped onto the structure of CLIC5 (PDB 6Y2H). **d** The difference in deuteration levels at the indicated time points between CLIC5-WT and CLIC5-F34D at pH 7.5 (upper panel) and pH 5.5 (lower panel). **(e)** The HDX difference between CLIC5-WT and CLIC5-F34D at the final time point is mapped onto the structure of CLIC5 (PDB 6Y2H).

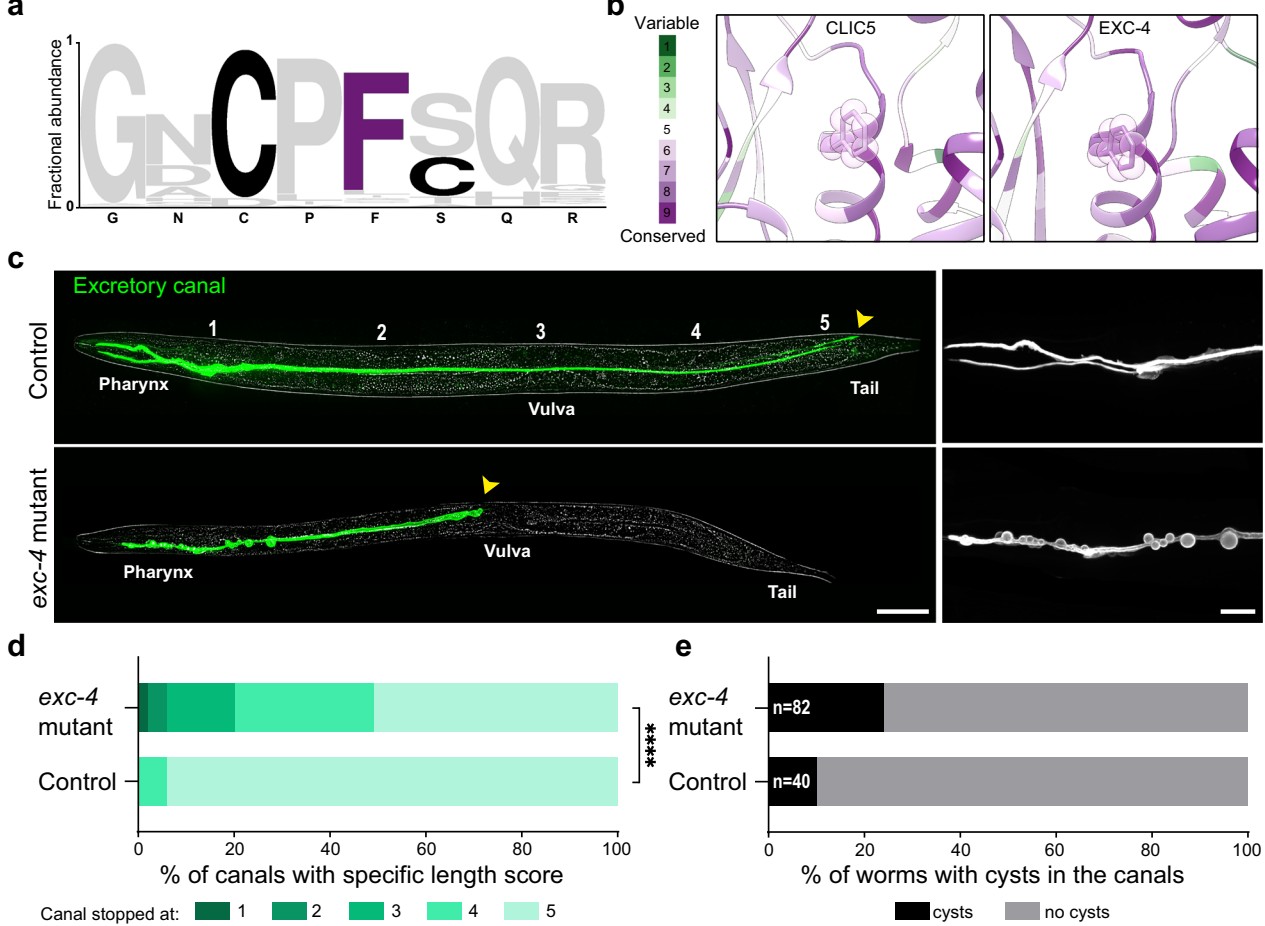

**Fig. 7 | Excretory canal defects in the nematode *C. elegans* with point mutation F38D in the *exc-4* gene. a** Logo plot of the fractional abundance of residues flanking CLIC5-F34D among human CLIC1-6 and *C. elegans* EXC-4. **b** Closeup view of CLIC5 F34 (left) and EXC-4 F38 (right), colored according to Consurf conservation score analyses. **c** Representative images of L4 stage worms expressing an excretory canal fluorescent marker (green). Control worm (upper panel) with an excretory canal extending from the pharynx to the tail, and *exc-4* mutant worm (lower panel) with an excretory canal extending only to the vulva and harboring visible cysts along the pharyngeal region are shown. Yellow arrows indicate the end of the canal in each worm. Magnified pharyngeal regions are presented on the right. Bars = 50 µm (left panels) and 20 µm (right panels). **d** Outgrowth scores of control vs. *exc-4* mutant worms. Significance was calculated with the two-sided Mann-Whitney U test, ****P < 0.0001. **e** Percentage of worms with cysts along the excretory canal in control and *exc-4* mutant worms.

outgrowth to the tail. The *exc-4* F38D mutant worms exhibited a significant number of excretory canals that did not reach their destination at the tail (Fig. 7d). In addition, 24% of mutant worms had small but visible cysts along the canal, while only 10% of control worms showed the same phenotype (Fig. 7e). CLIC5 in placental microvilli was found to interact with ezrin[21], whose worm ortholog, ERM-1, is known to be essential for excretory canal cell lumen extension[50]. To test whether the F38D mutation in EXC-4 affects ERM-1 localization, we introduced the *exc-4* F38D mutation in an ERM-1::GFP reporter strain[51]. As shown in Supplementary Fig. 10, the *exc-4* F38D mutation did not affect ERM-1 localization in the excretory canal, where it appeared as two parallel lines lining the lumen, similar to the control. Together, the *exc-4* F38D mutant features defective membrane remodeling in vivo, further supporting the physiological relevance of the fusogenic activity of CLICs.

## Discussion

Since their discovery, and despite extensive functional and structural studies, our understanding of CLICs' physiological role is incomplete. The CLIC family is highly conserved among metazoa, with most organisms expressing several isoforms/splice variants in a tissue-specific manner[1]. Historically, the discovery of CLICs led to their classification as a family of chloride channels[3,4,52]. However, the hypothesis

of CLICs' ability to function as ion channels in the physiological context is controversial[5]. While CLICs were repeatedly shown to mediate poorly specific ion conductance using artificial membranes, facilitated by oxidative environment and low pH[1], unequivocal evidence for the ability of the membrane-bound state of CLICs to function as chloride channels is still missing.

Purified proteins lacking ion channel activity were previously shown to mediate similar currents using artificial membranes[5]. Indeed, the classification of p64 protein (now known as the atypical long isoform CLIC5B) as a chloride channel was performed using this technique[3,4]. However, after reconstitution of the purified proteins into artificial membranes, the currents were unaffected by chloride channel inhibitors, including IAA-94[3], originally used to isolate this protein. Furthermore, immunoprecipitated CLIC5 failed to induce chloride channel activity[4]. Moreover, while CLICs-related conductance has been recorded following ectopic expression or from native membranes, it remains unclear whether their channel-forming activity is endogenous or secondary in nature[5,53,54]. These findings raised questions regarding the ability of CLICs to form anion channels at all [5]. Alternatively, CLICs may serve as an auxiliary subunit for a yet-to-be-determined pore-forming protein or give rise to a different type of membrane remodeling functionality.

The hypothesis that CLIC5 exhibits fusogen-like properties materialized via support from different lines of experimental evidence.

First, incubation of CLIC5 with liposomes resulted in increased opacity (Fig. 1a), due to a direct interaction between CLIC5 and the liposomes (Fig. 1b, c), leading to an increase in the liposomal diameter, as determined by DLS (Fig. 2). Next, we demonstrated that the increased diameter reflects the full-fusion of liposomal membranes, using fluorometric lipid- (Fig. 3) and content mixing (Fig. 4) assays. Moreover, no lipid scrambling nor content leakage, which could give rise to similar fluorescence changes, could be observed (Figs. 3d and 4c, respectively). Finally, the fusion activity is facilitated by acidic pH (Fig. 3b), previously shown to enhance the ability of CLIC1 to interact with membranes[10,55].

With the dramatic effect of acidic pH on the fusogenic activity of CLIC5, we hypothesized that the exposure of CLIC5 to such conditions would result in profound conformational changes. HDX-MS analysis confirmed that CLIC5 undergoes significant and extensive structural changes upon acidification. Specifically, deuterium uptake is markedly increased at the inter-domain interface under acidic conditions, indicating exposure of widespread regions to the bulk (Fig. 6b,c). Strikingly, we previously observed a similar effect following the exposure of CLIC5 to oxidative conditions, a known facilitator of CLICs' interaction with membranes[5,13]. Furthermore, we demonstrated that CLIC family members exhibit inherent flexibility in solution, with a minor population transitioning between compact and elongated conformations under reducing conditions at neutral pH[11]. Thus, our pH-dependence HDX-MS analyses support a population shift mechanism, skewing the conformational distribution towards the elongated form. Importantly, the transition to the elongated conformation involves the exposure of the hydrophobic inter-domain interface to the bulk (Supplementary Fig. 7), providing a mechanistic paradigm for the triggering event necessary for membrane interaction, initiating membrane fusion. To further explore the significance of inter-domain interface hydrophobicity for membrane fusion, we studied the CLIC5-F34D mutant. Notably, this phenylalanine residue is strictly conserved among CLICs, and mutation of the homologous position in CLIC4 prevented its translocation to the plasma membrane[29]. Consistent with our suggested mechanism, this mutant is recalcitrant to an acidic environment, interacts less with membranes, and lacks the ability to induce their fusion (Figs. 5 and 6).

Membrane fusion is crucial for a myriad of biological events[56]. Despite the immense diversity of membrane fusion functionalities, the fusion reaction itself is generic, relying on bringing together opposed membranes into high proximity to lower the free energy barrier needed for the fusion process to occur. Notably, the proposed sequence of events leading to CLIC5-mediated fusion is reminiscent of the influenza virus Hemagglutinin (HA), one of the most studied fusion proteins. Specifically, HA2 (subunit 2 of HA) orchestrates the fusion between the viral and the endosomal membrane. Importantly, HA2-mediated membrane fusion is induced by endosomal acidification, resulting in the exposure of a hydrophobic fusion peptide, which anchors to the endosomal membrane. Then, several HA2 proteins cluster to form a fusogenic unit, ultimately leading to membrane destabilization and fusion between viral and host membranes[57].

Similarly, we propose that acidification or oxidative environment leads to the exposure of the hydrophobic inter-domain interface of CLIC5, promoting membrane association, clustering[11,13], and ultimately membrane fusion (Fig. 8).

Transitioning to the physiological realm, CLICs were previously shown to be involved in vital cellular processes, sharing a common reliance on the occurrence of membrane remodeling and fusion events[17,20]. To link the inability of CLIC5-F34D to induce membrane fusion and the possible physiological significance of this family, we show that introduction of the homologous mutation into *C. elegans* EXC-4 results in excretory canal malformation reminiscent of, although not as severe, as complete knockout (Fig. 7). This is expected given the residual fusion activity of CLIC5-F34D (Fig. 5). Recently *exc-4*–mediated tubulogenesis was suggested to involve Gα-encoding genes in the Rho/Rac-signaling pathway[49]. Curiously, we show that CLIC5 is self-sufficient for promoting membrane fusion (Figs. 2–4). Nevertheless, *exc-4*–C237Y causes shortening of the excretory canal without the formation of large cysts, in contrast with the null phenotype and our *exc-4*–F38D mutant. Thus, excretory canal formation may rely on a complex multi-step process involving both direct *exc-4*–mediated membrane fusion and signaling via protein-protein interactions.

Together, our structural and functional investigations put forward the following mechanistic model for CLICs' function as fusogens (Fig. 8). At rest, most of the CLIC population exists in the globular conformation, in which the hydrophobic inter-domain interface is concealed from the bulk, an unfavorable state for membrane interaction. However, triggering events, such as acidic pH or exposure to reactive oxygen species (ROS), induce a population shift towards the elongated conformation, in which the hydrophobic inter-domain interface is exposed and can be utilized for downstream association with the membrane. Then, membrane-associated CLIC subunits undergo assembly into oligomeric complexes[11,13], resulting in the approximation and destabilization of adjacent membranes, overcoming the energetic barrier needed for full fusion to occur.

In summary, while the physiological roles of CLICs remain a matter of active research, the data we present provides a mechanistic framework to explore the involvement of CLICs in mediating membrane fusion events during numerous cellular processes.

## Methods

### Cloning and protein expression
Human CLIC5 (Uniprot Q9NZA1; CLIC domain (residues 16-251; termed CLIC5)), human CLIC2 (Uniprot O15247; residues 2-247), mouse CLIC6 (Uniprot Q8BHB9; residues 363-596), EXC-4 (Uniprot Q8WQA4; residues 2-290), rat 14-3-3ε (Uniprot P62260) and 14-3-3θ (Uniprot P68255) were sub-cloned into a pETM11 vector. A GAMG cloning artifact sequence, originating from the remanent TEV cleavage site and a NcoI restriction site, remained at the amino terminus. The F34D mutation was introduced using the standard QuickChange approach, and deletion of residues 57-68 was achieved by using the NEBaseChanger® workflow. The rat DOC2B (a kind gift from Prof. Daniel Khananshvili, Tel Aviv University; Uniprot P70610; residues 117-412)

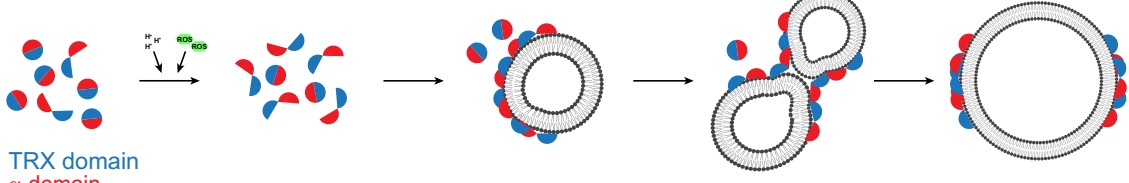

**Fig. 8 | Model for CLICs-induced membrane fusion.** CLICs are schematically represented as circles, composed of the TRX (blue) and α (red) domains. A triggering event (e.g., acidic pH or introduction of ROS) leads to increased exposure of the inter-domain interface, favoring membrane interaction and, finally, fusion. Created with elements adapted from BioRender.com.

was sub-cloned into pRSFduet vector. All constructs were verified by sequencing. All proteins were expressed in T7 Express *E. coli* cells (New England Biolabs) as previously described in ref. 13. Briefly, bacterial cultures were grown to mid-log phase in terrific broth (Formedium) at 37 °C and induced with 0.3 mM isopropyl β-D-1-thiogalactopyranoside (Formedium) overnight at 16 °C. Cells were harvested by centrifugation at 5500 x *g* for 15 min at 4 °C, and cell pellets were stored at −80 °C until use.

### Protein purification

All proteins were subjected to a similar procedure with slight modifications. To purify CLIC5, CLIC2, CLIC6, or EXC-4, a frozen bacterial paste was resuspended in CLIC buffer containing 150 mM NaCl, 50 mM Tris-HCl, and 1 mM tris(2-carboxyethyl)phosphine (TCEP), pH 8.0, supplemented with 15 mM imidazole and containing 2.5 μg ml⁻¹ DNaseI, 10 mg lysozyme (Fisher Scientific), and 2 mM phenylmethane sulfonyl fluoride (Acros Organics). Cells were lysed with an EmulsiFlex C-3 homogenizer (Avestin) and the lysate was cleared by centrifugation at 40,000 x *g* for 45 min at 4 °C. The clarified supernatant was then loaded onto a Ni²⁺ affinity resin column (HiTrap HP, Cytiva), followed by a wash step with CLIC buffer containing 27 mM imidazole and eluted using CLIC buffer supplemented with 300 mM imidazole. The hexahistidine tag was cleavaged overnight at 4 °C by TEV protease, followed by desalting (HiPrep 26/10, Cytiva) into CLIC buffer supplemented with 15 mM imidazole. The cleaved protein was re-loaded onto a Ni²⁺ column, and the flow-through was collected, concentrated to 4–5 mL, and subjected to size-exclusion chromatography (SEC) using a preparative-scale column (HiLoad 16/60 Superdex 75, Cytiva), equilibrated with gel filtration buffer, containing 150 mM NaCl, 20 mM Na⁺-HEPES, pH 7.5. Finally, pooled fractions were concentrated to ~200–1500 μM, depending on downstream experiments, using a 10 kDa molecular weight cut-off concentrator (Millipore), flash-frozen in liquid N₂, and stored at −80 °C until use. For 14-3-3ε and 14-3-3θ, the purification steps were identical to CLIC5 purification, except for the use of a HiLoad 16/600 Superdex 200 column (Cytiva) for gel filtration.

DOC2B purification was performed as previously described in ref. 58. Briefly, frozen cells were resuspended in 300 mM NaCl, 50 mM TRIS-HCl, pH 6.8 (DOC buffer), supplemented with 15 mM imidazole, 10 mg lysozyme, 1 mM phenylmethane sulfonyl fluoride, and 0.1% (w/v) Triton X-100 (Fisher BioReagents), followed by lysis, clarification, and Ni²⁺-mediated initial purification as above. Imidazole was removed using a desalting column pre-equilibrated with DOC buffer, supplemented with 5% (w/v) glycerol. Following the removal of the hexahistidine-tag using TEV (4 °C, overnight), the cleaved protein was re-loaded onto a Ni²⁺ column and eluted using DOC buffer, supplemented with 5% (w/v) glycerol and 50 mM imidazole. Finally, this eluate was collected, concentrated with a 10 kDa molecular weight cut-off concentrator, and loaded onto a preparative-scale column (HiLoad 16/60 Superdex 75, Cytiva), pre-equilibrated with 200 mM NaCl and 20 mM TRIS-HCl, pH 6.8. Pooled fractions were concentrated, flash-frozen in liquid N₂, and stored at −80 °C until use.

### Liposomes preparation

A mixture of 2.75:1.25:1 mg of chloroform-dissolved 1-Palmitoyl-2-Oleoyl-sn-Glycero-3-Phosphoethanolamine (POPE): 1-Palmitoyl-2-Oleoyl-sn-Glycero-3-Phosphoglycerol (POPG):cholesterol (Anatrace) were dissolved and mixed in chloroform, later removed by evaporation. The resulting dried lipid film was then resuspended to 10 mg/ml in assay buffer, containing 150 mM NaCl, 1 mM EDTA, and 50 mM Na₂HPO₄-NaH₂PO₄, pH 6.0 (for experiments conducted with DOC2B, EDTA was excluded). The lipid mixture was then sonicated in a bath sonicator for 15 min, followed by extrusion through a 0.2 μm membrane (Whatman) for 11–13 times (Avanti mini-extruder), to form homogeneous, large unilamellar vesicles (LUVs), which were stored at 4 °C until use.

### Dynamic light scattering (DLS)

DLS measurements were carried out using the Mobius instrument (Wyatt Technology), in a disposable 1 mL cuvette at a final volume of 700 μl. For CLIC5 and 14-3-3θ, LUVs (250 μM of lipids) were mixed with 25 μM protein in an assay buffer containing 150 mM NaCl, 1 mM EDTA, and 50 mM Na₂HPO₄-NaH₂PO₄, pH 6.0. As a control, CLIC5 was replaced with its gel-filtration buffer. DOC2B experimental procedures were performed using the respective DOC2B gel filtration buffer, containing 200 mM NaCl, 20 mM TRIS-HCl, pH 6.8. The samples included 2.5 μM DOC2B, 25 μM lipids, and 100 μM CaCl₂. As a control, CaCl₂ or DOC2B were substituted with 1 mM EDTA or gel filtration buffer, respectively.

### FRET experiments

All fluorescence spectroscopy experiments were performed in triplicates using a RF-8500 spectrofluorometer (Jasco) in assay buffer, containing 150 mM NaCl, 50 mM Na₃PO₄, pH 5.5. The occurrence of FRET was monitored between native tryptophan residues in CLIC5 or DOC2B and dansyl-PE (phosphatidylethanolamine)-containing liposomes. For liposome preparation, the lipid mixture was supplemented with 2% (w/v) dansyl-PE (Invitrogen), followed by LUVs preparation as described in the liposomes preparation method section. FRET experiments between the proteins (10 μM) and lipids (2500 μM) were performed using 290 nm and 340–550 nm excitation and emission wavelengths, respectively, and the reaction was monitored periodically over the indicated time course at room temperature.

### Thermal-shift assay (TSA) analysis

TSA[47] was performed using a real-time PCR system (StepOne Plus, Life Technologies), with the fluorescent dye SYPRO Orange (Invitrogen), utilizing the carboxyrhodamine (ROX) filter set in clear 96 well plates. The temperature was increased using a continuous ramp at a 1 °C/minute rate from 4 °C to 95 °C. Assays were performed in a 20 μl final volume containing 4 μM protein in reaction buffer, 5X SYPRO Orange, 150 mM NaCl, 20 mM Na-HEPES, pH 7.5.

### R18-based lipid mixing assay

Non-fluorescent liposomes were prepared as described above. For R18-labeled liposome preparation, the lipid mixture was supplemented with 5 mol% of R18 (100 mg/ml in DMSO; Invitrogen). Fluorometry experiments were performed using a FP-8500 spectrofluorometer (Jasco) in a 96-well black plate, with a final reaction volume of 200 μl. The excitation/emission wavelengths were 560/580 nm, respectively, and 5 nm excitation and emission slits were used. The final concentration of lipids used in the experiments was 250 μM with a 1:4 ratio between R18-labeled and non-fluorescent liposomes, respectively. For background subtraction, the protein used in the experiment was substituted with its respective gel-filtration buffer and subjected to the same experimental procedure.

### Carboxyfluorescein-based content mixing assay

Non-fluorescent liposomes were prepared as described above. 5(6)-Carboxyfluorescein (CF) (Merck; 100 mM CF stock, dissolved in 300 mM NaOH) was mixed in a 1:1 ratio with a buffer, containing 300 mM NaCl, and 100 mM Na₂HPO₄-NaH₂PO₄, pH 6. The dried lipid film was rehydrated with the 50 mM CF-stock solution, followed by sonication and extrusion. Finally, to remove the excess CF, the liposomes were applied onto a gravity flow desalting column (PD MiniTrap G25, Cytiva), pre-equilibrated with an elution buffer containing 150 mM NaCl, and 50 mM Na₂HPO₄-NaH₂PO₄, pH 6. Liposome content mixing fluorometry measurements were performed using a Varioskan LUX Fluorometer plate reader (ThermoFisher Scientific), similar to the R18-based assay in a 96-well format (200 μL reaction volume). Briefly, 490/515 nm excitation/emission wavelengths and 5 nm slits were used. The liposome mixture consisted of a 1:2 ratio between CF-

encapsulating and empty LUVs (250 μM total lipid concentration), respectively, and measurements were performed in the elution buffer.

## Scrambling and liposome leakage assays

For the scrambling assay[43], NBD-labeled liposomes were prepared by adding 1% (w/v) nitrobenzoxadiazole (NBD)-labeled lipid (NBD-PE) to the lipid mixture. The scrambling assay was performed in a 100 μl quartz cuvette at room temperature (250 μM final lipid concentration). The NBD-labeled liposomes were incubated with CLIC5 (25 μM), and the NBD fluorescence was monitored before and following the addition of dithionite (to 5 mM) using 460/538 nm excitation/emission, respectively (Jasco FP-8500 spectrofluorometer). Additional dithionite (5 mM) and Triton X-100 (0.5% w/v) were added to dissolve the liposomes and allow complete quenching of the NBD fluorescence.

For the NBD-glucose leakage control assay[43], the lipid film was equilibrated with NBD-glucose-containing buffer (12.6 μM). Excess NBD-glucose was removed by passing the liposomes through a desalting column (PD MiniTrap G25, Cytiva) prewashed with NBD-glucose-free buffer. Next, the experimental procedure was performed similarly to the scrambling assay, with first dithionite addition (to 1 mM), following second dithionite addition and (1 mM) and Triton X-100 (0.5% w/v) to quench the NBD-glucose signal completely.

## Liposome co-floatation assay

Liposome co-floatation assay was carried out using a previously established procedure[59]. Briefly, R18-labeled liposomes (250 μM lipids) were incubated with CLIC5, DOC2B, or 14-3-3ε (25 μM) at 25 °C in assay buffer, containing 150 mM NaCl, and 50 mM $Na_2HPO_4$-$NaH_2PO_4$, pH 5.5. After 90 minutes, 60% (w/v) sucrose was added (40 % final concentration), and the samples were transferred to 2.2 ml ultracentrifuge tubes. Then, each sample was overlaid with 450 μL of 30% (w/v) sucrose and 450 μL of assay buffer on the top. All sucrose solutions were prepared in the assay buffer. After centrifugation at 150,000 x $g$ for 90 min in a Beckman TLS-55 rotor, 13–15 μl fractions were collected throughout the gradient and analyzed by SDS-PAGE and fluorometry (detailed method in the R18-based lipid mixing assay section).

## Protein crystallization and data collection

Crystals of CLIC5-WT and F34D (Δloop) were grown at 19 °C using hanging-drop vapor diffusion by mixing a 1:1 ratio (v/v; Mosquito, TTP Labtech) of protein solution at 1858 μM (CLIC5-WT) and 1913 μM (CLIC5-F34D) and a reservoir solution, containing 0.2 M Ammonium sulfate 0.1 M BIS-TRIS pH 7.0 30% Polyethylene glycol 3350. For diffraction data collection, crystals were immersed in liquid $N_2$ after cryoprotection with 20% glycerol. Data were collected at 100 K on beamline ID30B of the European Synchrotron Radiation Facility (ESRF), using a wavelength of 0.976 Å. Integration, scaling, and merging of the diffraction data were done with the XDS program[60]. The crystals belonged to space group $C222_1$ (Table 1).

## Structure determination and refinement

Structures were solved by automated molecular replacement using Phaser[61] with the structure of human CLIC5-WT (Protein Data Bank (PDB) 6Y2H, chain C) as a search model. Iterative model building and refinement were carried out in Phenix[62] with manual adjustments using Coot[63]. Ramachandran analysis was performed using MolProbity[64]. Data collection and refinement statistics are presented in Table 1. Structural illustrations were prepared with UCSF Chimera (www.cgl.ucsf.edu/chimera) and PyMOL molecular graphics system version 2.0.6 (Schrödinger, LLC).

## Molecular dynamics (MD) simulations

All the simulations were performed using the Schrödinger Maestro release 2022-3 (Schrödinger LLC). First, missing loops were added to

## Table 1 | Crystallographic statistics

| Data collection | | |
| --- | --- | --- |
| | CLIC5-Δloop-WT | CLIC5-Δloop-F34D |
| Spacegroup | $C222_1$ | $C222_1$ |
| Cell dimensions | | |
| a,b,c (Å) | 130.13, 220.83, 45.02 | 129.69, 220.29, 45.48 |
| α,β,γ (°) | 90, 90, 90 | 90, 90, 90 |
| Beamline | ESRF ID30B | |
| Wavelength (Å) | 0.886 | 0.886 |
| Resolution (Å) | 2.10 | 2.51 |
| Multiplicity | 16.7 (17.1) | 6.5 (6.8) |
| Completeness (%) | 99.8 (99.5) | 99.6 (98.6) |
| Mean I/σ(I) | 19.1 (1.4) | 11.8 (1.1) |
| $R_{meas}$ (%) | 8.5 (184.3) | 12.6 (197.5) |
| $CC_{1/2}$ (%) | 100.0 (85.1) | 99.8 (50.3) |
| **Refinement statistics** | | |
| No. reflections (work/free) | 36476/2005 | 21694/1141 |
| Resolution range | 42.56–2.10 | 42.42–2.51 |
| $R_{work}$/$R_{free}$ | 0.2321/0.2525 | 0.2438/0.2730 |
| No. atoms | | |
| Macromolecules | 3209 | 3205 |
| Ligands | 10 | 5 |
| Solvent | 49 | 25 |
| average B-factor | | |
| Macromolecules | 74.8 | 79.8 |
| Ligands | 100.9 | 115.6 |
| Solvent | 58.9 | 64.2 |
| RMSD (bond lengths) | 0.005 | 0.005 |
| RMSD (bond angles) | 0.71 | 0.74 |
| Ramachandran outliers (%) | 0 | 0 |

Values in parentheses are for the highest-resolution shell. *RMSD* Root mean square deviations.

the structure of CLIC5-WT (PDB 6Y2H) using MODELLER[65], and the F34D mutation was introduced using Schrödinger Maestro release 2022-3 (Schrödinger, LLC). The structure was prepared using the Protein Preparation Wizard. Missing hydrogen atoms were added considering a pH value of 7.4 ± 1.0 or 5.5 ± 1.0, followed by optimization of the hydrogen bond network. Next, energy minimization was performed using MacroModel (Schrödinger Release 2022-3: MacroModel) with the OPLS4 forcefield and Polack-Ribiere Conjugate Gradient (PRCG) algorithm. The system setup tool was used to solvate the systems using the TIP3P solvent model. Potassium or chloride ions were added to neutralize the charge and to obtain a final salt concentration of 150 mM (Supplementary Table 1). All MD simulations were performed using Desmond with the OPLS4 force field[66]. The simulations were conducted under a Langevin temperature and pressure control, using periodic boundary conditions with particle-mesh Ewald (PME) electrostatics with a 12 Å cutoff for long-range interactions. The systems were equilibrated using the default relaxation protocol, and finally, three all-atom production simulations for each protein were carried out for 1 μs with a constant pressure of 1 atm and a constant temperature of 300 °K starting from a random seed. The results were manually inspected and analyzed using the Maestro suite.

## Hydrogen/deuterium exchange mass spectrometry (HDX-MS)

The H/D exchange reactions were prepared using a PAL DHR autosampler (CTC Analytics AG) controlled by Chronos software (Axel-Semrau). Protein at the final concentration of 20 μM in 150 mM NaCl, 50 mM $Na_3PO_4$ with pH adjusted to 7.5 or 5.5 (assay buffer),

respectively, was diluted 10-fold by the corresponding D$_2$O-based assay buffer of pD 7.5 and 5.5. HDX was followed for 20 s, 60 s, 300 s, 2000 s and 6000 s intervals. Time-points 20 s and 300 s were triplicated. The exchange reaction was quenched by the addition of chilled 1 M glycine-HCl pH 2.3 at 1:1 (v/v) ratio. Samples were immediately injected onto the LC system placed in the Peltier-cooled box, and coupled to Agilent Infinity II 1260/1290 UPLC (Agilent Technologies) and an ESI source of timsTOF Pro equipped with PASEF (Bruker Daltonics). The LC setup consisting of custom-made pepsin/nepenthesin-2 column (bed volume 66 μL), trap column (SecurityGuard™ ULTRA Cartridge UHPLC Fully Porous Polar C18, 2.1 mm ID; Phenomenex) and an analytical column (Luna Omega Polar C18, 1.6 μm, 100 Å, 1.0 × 100 mm; Phenomenex) was cooled to 0 °C to minimize the back-exchange. Proteins were digested, and peptides desalted by 0.4% formic acid (FA) in water delivered by the 1260 Infinity II Quaternary pump at 200 μL/min$^{-1}$. To elute and separate desalted peptides, water-acetonitrile (ACN) gradient (10–45%; solvent A: 0.1% FA in water, solvent B: 0.1% FA, 2% water in ACN) followed by a step to 99% B was used, and the solvents were driven by the 1290 Infinity II LC system pumping at 40 μL/min$^{-1}$. Mass spectrometer operated in the MS mode with 1 Hz data acquisition rate. Fully deuterated controls were prepared for WT and F34D variant. Acquired LC-MS data were peak picked and exported in DataAnalysis (v. 5.3, Bruker Daltonics) and further processed using DeutEx software[67]. Data visualization was performed using MSTools (http://peterslab.org/MSTools/index.php)[68] and PyMOL molecular graphics system version 2.0.6 (Schrödinger, LLC). For peptide identification, the same LC-MS system as described above was used, but the mass spectrometer was operated in data-dependent MS/MS mode with PASEF active and tims enabled. The LC-MS/MS data were searched using MASCOT (v. 2.7, Matrix Science) against a custom-built database combining a common cRAP.fasta (https://www.thegpm.org/crap/) and the sequences of CLIC5-WT, -F34D and used proteases. Search parameters were set as follows: precursor tolerance 10 ppm, fragment ion tolerance 0.05 Da, decoy search enabled, FDR ‹1%, Ion-Score ›20 and peptide length ›5.

### *C. elegans* strains and handling
Strains used in this study: N2 – wild type; SYS1728 - mir-232p::mNeonGreen::PH::tbb-2 3′UTR + NeoR with lineage reporter (provided by Zhuo Du, Chinese Academy of Sciences); RZB452 - exc-4 (msn213[F80D]); mir-232p::mNeonGreen::PH::tbb-2 3′UTR + NeoR with lineage reporter. All *C. elegans* strains were grown and maintained on nematode growth medium (NGM) plates seeded with OP50 bacteria according to standard protocols (Brenner, 1974). All worms were kept at 20 °C.

### Generating the *exc-4* mutant F38D strain
For the generation of *exc-4* mutant worms, we used the CRISPR-Cas9 method of genome editing (Paix et al., 2017). A sgRNA guide was selected with a PAM site 16 bp away from the codon of phenylalanine 38 in the *exc-4* gene sequence (antisense strand:5′AGAAAAGA TCGGCTCCAATG3′). An HDR repair template (single stranded oligodeoxynucleotide; ssODN) was designed with two mutations to change amino acid 38 from phenylalanine to aspartic acid, as well as three more silent mutations to prevent reattachment and recutting by Cas9 and a 35nt homology arms flanking each side (AAAGAGCATA-CAACTCCATCCAGAATTCCTGACAGTCAAGATCTGCGCCAATGCGG CGAGCATCAATTCCTGACGCTTTTAC). In addition, the introduction of silent mutations created a new restriction site for BglII restriction enzyme, which facilitated genotyping by PCR. The injection mix included Cas9 protein (0.8 μg/μl) (Integrated DNA Technologies), *exc-4* sgRNA (0.06 mM), ssODN (0.225 μg/μl), *dpy-10* sgRNA (0.02 μg/μl), *dpy-10* ssODN (0.04 μg/μl), KCl (25 mM) and tracrRNA (0.1 μg/μl). This injection mix was injected into the gonads of young adult stage N2 worms. F1 progeny were screened for phenotypically dumpy worms. These F1 dumpy worms were isolated, allowed to lay embryos, and later screened for *exc-4* F38D mutation by PCR using following primers: forward primer TGTAGTGGTACAATAGTGGTACGG and reverse primer AGTTCTCTATTCTCTTCTCAGCGG, and cutting the PCR product with BglII. The mutations were further confirmed by sequencing. Homozygous mutant worms were then crossed into the SYS1728 reporter strain to receive mutant worms with a fluorescent excretory canal. All CRISPR products and primers were purchased from IDT (Integrated DNA Technologies).

### Microscopy and image acquisition
Live L4 hermaphrodites in 10 mM levamisole were mounted on fresh 3% agarose pads prepared on a glass slide. Imaging was carried out with a Nikon Ti2E microscope equipped with a Yokogawa W1 spinning-disc system and a Plan Apo 60× oil 1.4 NA and a Plan Apo 20 × 0.75 NA. Samples were illuminated with 488 nm laser (Gataca systems) for excitation and acquired on a Prime 95B sCMOS camera (Photometrics). The software MetaMorph version 7.10.2.240 (Molecular Devices) was used as the controlling interface. All images were captured with z stacks of 0.4 μm spacing.

### Reporting summary
Further information on research design is available in the Nature Portfolio Reporting Summary linked to this article.

## Data availability
The data that support this study are available from the corresponding authors upon request. The atomic coordinates have been deposited in the Protein Data Bank (PDB) under accession codes 8Q4I (CLIC5-Δloop); and 8Q4J (CLIC5-Δloop-F34D). The mass spectrometry proteomics data have been deposited to the ProteomeXchange Consortium via the PRIDE[69] partner repository with the dataset identifier PXD045186. All source data for Figs. 1–5 and uncropped gel images corresponding to Figs. 1b,c and 5a are provided as a source data file. The initial configuration and the final frame of each MD simulation are provided as a Source Data file. Source data are provided with this paper.

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

## Acknowledgements
We thank the staff of ID30B at the European Synchrotron Radiation Facility (ESRF) for assistance with diffraction experimentation and Dr. Reuven Wiener for technical assistance with diffraction data collection. This work was performed in partial fulfillment of the requirements for a Ph.D. degree of B.M., Faculty of Medicine, Tel Aviv University, Israel. This work was supported by the Israel Science Foundation (grant numbers 1721/16 and 1653/21 (Y.H.) and 3308/20 (R.Z.B)), the Israel Cancer Research Fund grants 01214 and (Y.H.) and 19202 (M.G.), the Israel Cancer Association grants 20230029 (Y.H. and M.G.), and the Kahn Foundation's Orion project, Tel Aviv Sourasky Medical Center, Israel (M.G.). Support also came from the Claire and Amedee Maratier Institute for the Study of Blindness and Visual Disorders, Faculty of Medicine, Tel-Aviv University (Y.H. and M.G.). Access to MS installation was funded by the EU Horizon 2020 grant EU_FT–ICR_MS project number 731077 and by CIISB (LM2023042). P.M. and P.V. support from MEYS CZ funds CZ.1.05/1.1.00/02.0109 is gratefully acknowledged.

## Author contributions
Conceptualization, M.G. and Y.H.; Methodology, R.Z.B., P.M., M.G. and Y.H.; Investigation, B.M., A.V., P.V., A.N., P.M., M.G., and Y.H.; Formal Analysis, B.M., P.M., A.N., M.G., and Y.H.; Writing – Original Draft, M.G. and Y.H.; Writing – Review and Editing, B.M., A.V., P.V., A.N., R.Z.B., P.M., M.G., and Y.H.; Supervision, M.G. and Y.H.; Funding Acquisition, R.Z.B., P.M., M.G., and Y.H.

## Competing interests
The authors declare no competing interests.
