## [Peer Review File · Nature Communications]

Chloride intracellular channel (CLIC) proteins function as fusogensReviewer #1 (Remarks to the Author):

This is an exciting paper. The CLIC proteins were discovered nearly 30 years ago, they are highly conserved in metazoa, however, understanding their function has proved very difficult. The metamorphic nature of the CLIC proteins and their transition from soluble proteins to membrane-bound proteins have made functional studies challenging.

This paper provides new insight into the interaction of hCLIC5 with membranes and it is likely to impact our understanding of all the CLIC proteins. The experimental data presented clearly shows that hCLIC5 binds to liposomes and facilitates liposomal fusion. hCLIC5 mediated liposomal fusion is complete with both lipids and liposomal content mixing. However, hCLIC5 has no scramblase activity. The data as well presented and clearly explained.

A surprising result is that the F34D mutant greatly reduces the liposomal fusion function. This phenylalanine is within a highly conserved CPFSQR sequence motif and it lies on the surface of CLIC5, near the interface between the N-terminal Trx domain and the C-terminal helical domain. In CLIC4, this mutation was previously shown to prevent the transition of cytosolic CLIC4 to the plasma membrane after GPCR stimulation.

The paper characterises the F34D mutant in comparison to WT hCLIC5. Unsurprisingly, the crystal structures of WT and F34D hCLIC5 are essentially identical. However, the authors show that the dynamics of hCLIC5 in solution is drastically altered by the F34D mutation. In general, the CLIC-membrane interaction is enhanced by low pH and/or oxidative conditions. These authors have previously shown that both low pH and oxidation alter the ensemble characteristics of hCLIC5 in solution (1). While the crystal structure is globular, SAXS data has shown that in solution hCLIC5 exists in this globular state in equilibrium with an elongated form. Low pH or oxidising conditions enhance the population in the elongated state. This change in dynamics was previously shown to correlation with changes in hydrogen-deuterium exchange (HDX), particularly in residues around the interdomain boundary.

In this paper, HDX shows that the introduction of the F34D mutation maintains hCLIC5 in the more globular form even at low pH. Thus, this mutation is stabilising the globular form and hence preventing (or greatly reducing) membrane interaction.

Finally, they show that the equivalent mutation in the *C. elegans* CLIC called Exc4 produces a phenotype that is similar (though not as severe) to the Exc4 knockout. The current result implies a molecular mechanism for the Exc4 knockout phenotype.

Overall, this is an exciting discovery for the CLIC field and for understanding metamorphic proteins that transit between soluble and membrane-bound states.

My only disappointment with this paper is that they do not address how the F34D mutation stabilises the globular form. Phe34 sits in a slot along the boundary between the Trx and the helical domains. One would expect this residue to enhance membrane interaction if the two domains separate to form a membrane-binding hydrophobic surface. One possible explanation for the stabilisation of the globular form is the presence of highly conserved basic residues straddling the interface: Arg37 and Lys191 – each neighbouring residue 34. Given that the authors have the mutant crystal structure, some analysis of this interface is warranted, given the increased stability of F34D particularly at low pH (thermal stability assay, SI Fig S5).

Minor points

1. Page 3, first sentence: CLICs are not highly conserved in eukaryotes. Instead, they are highly conserved in metazoa (2, 3). There exist families of related proteins in all three domains of life (3, 4), however, it is not clear that these non-metazoan proteins have the same function as metazoan CLIC

proteins. The reference, Gururaja Rao et al., 2018 never states that CLICs are highly conserved in eukaryotes (5).

2. Page 9: "Unexpectedly, despite the marked functional effect....no significant difference between WT and F34D." For a structural biologist, a point mutation rarely makes any significant difference to a crystal structure. The result observed by crystallography is entirely expected.

3. Page 1, second sentence of the Discussion: again, the CLIC family is not highly conserved among eukaryotes (see Minor Point 1). The reference given never makes such a statement.

4. Page 17, Materials & Methods: "A GAMG artifact sequence was introduced at the amino terminus." This is a bit strange and no explanation is given. This artifact was already present in their previous paper (1) reporting the structure of CLIC5. I note that the construct starts (after GAMG) with residue 16, thus removing the natural N-terminus. However, this region of the CLIC is unstructured (in all CLIC structures to date) and it is unlikely to alter the results presented. A little more explanation or reference would have been good.

5. References: I have not proofread the References, however, casual inspection has already picked up two errors: Ref 20 should be "Hobert" and not "Hoberst"; Ref 54 has no title.

6. Figure captions are a bit brief. For example, in Fig 3b, I have no idea what is being shown. I can see that fluorescence increases more rapidly at low pH, however, I have no idea as to the protein concentration nor any other experimental conditions.

7. Figure 6: in panel b, the right hand side says: "2000s (pH 5.5) -20s (pH 7.5)" on one line and "6000s (pH 5.5) -60s (pH 7.5)" on the next. Why these numbers? are the 20s and 60s significant? In contrast, panel d has "20s, 60s etc". Why the difference in reporting? Is it significant?

8. SI Fig S2: Something is not right here. Panel a looks like F34D (gold) superposed on the loop deletion mutant delta57-68. while Panel b looks like 6Y2H versus the WT deletion mutant. This is not what the caption says and it is not what the labels in the figure say.

9. SI Fig S3 panels b and c: why do you compare pH 7.5 at 60 seconds but pH 5.5 at 6,000 seconds? Is there a reason for using different time points? This seems strange.

10. SI Figure S5: these thermal shift assay plots look like the derivative of the fluorescence with respect to temperature (dF/dT) and not the normalised fluorescence as the axis labels say.

References

1. A. Ferofontov, P. Vankova, P. Man, M. Giladi, Y. Haitin, Conserved cysteine dioxidation enhances membrane interaction of human Cl(-) intracellular channel 5. *FASEB J* 34, 9925-9940 (2020).
2. L. Jiang et al., CLIC proteins, ezrin, radixin, moesin and the coupling of membranes to the actin cytoskeleton: a smoking gun? *Biochim Biophys Acta* 1838, 643-657 (2014).
3. D. R. Littler et al., The enigma of the CLIC proteins: Ion channels, redox proteins, enzymes, scaffolding proteins? *FEBS Lett* 584, 2093-2101 (2010).
4. S. Gururaja Rao et al., Identification and Characterization of a Bacterial Homolog of Chloride Intracellular Channel (CLIC) Protein. *Sci Rep* 7, 8500 (2017).
5. S. Gururaja Rao, D. Ponnalagu, N. J. Patel, H. Singh, Three Decades of Chloride Intracellular Channel Proteins: From Organelle to Organ Physiology. *Curr Protoc Pharmacol* 80, 11 21 11-11 21 17 (2018).

Reviewer #2 (Remarks to the Author):

The manuscript by Manori et al. provides novel information on the role of CLIC5 as a fusogen. CLICs are generally understudied proteins, and the functional role of these proteins is not yet elucidated. These proteins exist in membrane and soluble forms and both forms have distinct roles. Authors have shown that CLIC5 interacts with the membrane and induces fusion by using several independent approaches and the data is compelling. An amino acid F34 (possibly) in the transmembrane domain was shown to ablate this fusion. Experiments in *C. elegans* are supportive of in vitro findings in the in vivo model. There are several concerns for the manuscript which should be addressed.

1. The title states, "CLIC proteins function as fusogens". In the manuscript, the authors have only tested CLIC5 and not all the CLICs. The title does not reflect the outcome of the manuscript.
2. CLICs share homology with each other, and authors should also test whether fusogen property is exclusive for CLIC5 or is also shared by other CLIC proteins. This data will be highly beneficial for the manuscript along with DOC2B and 14-3-3.
3. The F34 residue is located in the putative transmembrane domain and is conserved in all the CLICs with the exception of CLIC3 (and Exl-1). CLIC3 could also serve as a control for the F34 claim.
4. The joint loop (57-68) is critical for the folding and insertion of CLICs. The data is convincing, but I would recommend using CLIC5 without a putative trans-membrane domain as it will not be able to insert into the membrane. In contrast, the CLIC5 1-68 construct should promote the fusogen role.
5. CLIC5 is known to interact with ezrin and other cytoskeletal proteins. In *C. elegans*, did authors observe any changes in the cytoskeletal proteins that are known to interact with CLIC5? This data should be presented.
6. The pH data is very exciting, however, the insertion of CLICs is also redox-dependent via a cysteine residue on top of the transmembrane domain, and it will be useful to see if the cysteine is also involved in the function.

Minor comments

1. There are some spelling mistakes such as 'wavelenght'
2. Usually POPE and POPS are used for CLICs. Please provide an explanation of why POPG is used as PE and PG are the main lipid components of the inner bacterial membrane.
3. In the cartoon, protein is shown on the outer leaf of the bilayer and not spanning the membrane. Is there a specific reason for not adding protein to both membranes (spanning)?

Reviewer #1:

This is an exciting paper. The CLIC proteins were discovered nearly 30 years ago, they are highly conserved in metazoa, however, understanding their function has proved very difficult. The metamorphic nature of the CLIC proteins and their transition from soluble proteins to membrane-bound proteins have made functional studies challenging.

This paper provides new insight into the interaction of hCLIC5 with membranes and it is likely to impact our understanding of all the CLIC proteins. The experimental data presented clearly shows that hCLIC5 binds to liposomes and facilitates liposomal fusion. hCLIC5 mediated liposomal fusion is complete with both lipids and liposomal content mixing. However, hCLIC5 has no scramblase activity. The data as well presented and clearly explained.

A surprising result is that the F34D mutant greatly reduces the liposomal fusion function. This phenylalanine is within a highly conserved CPFSQR sequence motif and it lies on the surface of CLIC5, near the interface between the N-terminal Trx domain and the C-terminal helical domain. In CLIC4, this mutation was previously shown to prevent the transition of cytosolic CLIC4 to the plasma membrane after GPCR stimulation.

The paper characterises the F34D mutant in comparison to WT hCLIC5. Unsurprisingly, the crystal structures of WT and F34D hCLIC5 are essentially identical. However, the authors show that the dynamics of hCLIC5 in solution is drastically altered by the F34D mutation. In general, the CLIC-membrane interaction is enhanced by low pH and/or oxidative conditions. These authors have previously shown that both low pH and oxidation alter the ensemble characteristics of hCLIC5 in solution (1). While the crystal structure is globular, SAXS data has shown that in solution hCLIC5 exists in this globular state in equilibrium with an elongated form. Low pH or oxidising conditions enhance the population in the elongated state. This change in dynamics was previously shown to correlation with changes in hydrogen-deuterium exchange (HDX), particularly in residues around the interdomain boundary.

In this paper, HDX shows that the introduction of the F34D mutation maintains hCLIC5 in the more globular form even at low pH. Thus, this mutation is stabilising the globular form and hence preventing (or greatly reducing) membrane interaction.

Finally, they show that the equivalent mutation in the *C. elegans* CLIC called Exc4 produces a phenotype that is similar (though not as severe) to the Exc4 knockout. The current result implies a molecular mechanism for the Exc4 knockout phenotype.

Overall, this is an exciting discovery for the CLIC field and for understanding metamorphic proteins that transit between soluble and membrane-bound states.

Response: We thank the reviewer for the thorough assessment and insightful comments.

My only disappointment with this paper is that they do not address how the F34D mutation stabilises the globular form. Phe34 sits in a slot along the boundary between the Trx and the helical domains. One would expect this residue to enhance membrane interaction if the two domains separate to form a membrane-binding hydrophobic surface. One possible explanation for the stabilisation of the globular form is the presence of highly conserved basic residues straddling the interface: Arg37 and Lys191 – each neighbouring residue 34. Given that the authors have the mutant crystal structure, some analysis of this interface is warranted, given the increased stability of F34D particularly at low pH (thermal stability assay, SI Fig S5).

Response: Thank you for this important comment. As indicated by the Reviewer, two potential electrostatic interactions may form with D34 – with R37 and/or K191. Close examination of CLIC5-F34D crystal structure revealed that only R37 is within a salt-bridge distance (3.7 Å) from D34, while K191 is more distant (5.7 Å). Importantly, the salt-bridge between D34 and R37 is not expected to stabilize the inter-domain interface, as it occurs between two adjacent residues, located in the same helix. Nevertheless, to better explore the dynamic nature of the interface, we performed 1 μsec molecular dynamics simulations of the CLIC5-F34D mutant at both pH 7.4 and pH 5.5, as discussed in the revised manuscript. These simulations reveal that a salt-bridge interaction forms between D34 and K191 over a substantial fraction of the simulation trajectories (new Supplemental Figure 8). Of note, this interaction is more stable at pH 7.4, consistent with the TSA results (Supplemental Figure 7). Thus, our newly added data indicates, as suggested by the reviewer, that the salt-bridge interaction contributes to the stability of the F34D mutant, at both pH values.

A section describing these findings was added to the results:

Page 10, last paragraph: “Close inspection of the CLIC5-F34D crystal structure reveals two positively charged residues, which can potentially form a salt-bridge with D34 (Fig. 6a). R37 is within a salt-bridge distance (3.7 Å), while K191 is not captured interacting with D34 in the static structure (Supplementary Figure 8a). Given the dynamic nature of the interface, we performed 1 μsec molecular dynamics simulations of the CLIC5-F34D mutant at both pH 7.4 and pH 5.5 (Supplementary Figure 8). All the simulations performed here have reached convergence, as reflected by the plateau in root mean square deviation (RMSD) values (Supplementary Figure 8b). In addition, the per-residue root mean square fluctuation (RMSF) values (representing its spatial fluctuation relative to its average position along the simulation) are similar at both pH values (Supplementary Figure 8c). These simulations reveal that a salt-bridge interaction forms between D34 and K191 over a substantial fraction of the simulation trajectories (Supplementary Figure 8d). Of note, this interaction is more stable at pH 7.4. In agreement with the HDX-MS analysis, the TSA analysis of this mutant showed very weak pH-dependent differences in the basal fluorescence signal (from 0.01 ± 0.01 , at pH 7.5, to 0.06 ± 0.01 , at pH 5.5; $P < 0.05$, $n = 6$), consistent with low exposure of cryptic hydrophobic surfaces (Supplementary Figure 7). However, CLIC5-F34D undergoes thermal denaturation at a lower temperature range when exposed to pH 5.5, consistent with the reduced stability of interfacial salt-bridges.”

Minor points

1. Page 3, first sentence: CLICs are not highly conserved in eukaryotes. Instead, they are highly conserved in metazoa (2, 3). There exist families of related proteins in all three domains of life (3, 4), however, it is not clear that these non-

metazoan proteins have the same function as metazoan CLIC proteins. The reference, Gururaja Rao et al., 2018 never states that CLICs are highly conserved in eukaryotes (5).

Response: Thank you for this comment. We have clarified this point in the revised manuscript as you suggested.

Page 3, first sentence: "The Chloride Intracellular Channel (CLIC) family is highly conserved among metazoa, where most organisms express several isoforms/splice variants in a tissue-specific manner."

2. Page 9: "Unexpectedly, despite the marked functional effect...no significant difference between WT and F34D." For a structural biologist, a point mutation rarely makes any significant difference to a crystal structure. The result observed by crystallography is entirely expected.

Response: Thank you for pointing that out. Given the hydrophobic nature of the interface, and the presence of neighboring positively charged residues, the introduction of a negatively charged residue could have distorted the local interaction network. However, in line with the Reviewer's suggestion, we omitted the word 'unexpextedly' from this sentence.

Page 9, second paragraph: "Despite the marked functional effect conferred by the F34D mutation, the static high-resolution structure of the compact conformation reveals no significant differences between WT and F34D."

3. Page 1, second sentence of the Discussion: again, the CLIC family is not highly conserved among eukaryotes (see Minor Point 1). The reference given never makes such a statement.

Response: Thank you. We corrected this statement as suggested.

Page 12, last paragraph: "The CLIC family is highly conserved among metazoa, with most organisms expressing several isoforms/splice variants in a tissue-specific manner."

4. Page 17, Materials & Methods: "A GAMG artifact sequence was introduced at the amino terminus." This is a bit strange and no explanation is given. This artifact was already present in their previous paper (1) reporting the structure of CLIC5. I note that the construct starts (after GAMG) with residue 16, thus removing the natural N-terminus. However, this region of the CLIC is unstructured (in all CLIC structures to date) and it is unlikely to alter the results presented. A little more explanation or reference would have been good.

Response: Thank you for highlighting this issue. We have now included a more detailed explanation regarding this cloning artifact.

Page 18, first paragraph: "A GAMG cloning artifact sequence, originating from the remanent TEV cleavage site and a NcoI restriction site, remained at the amino terminus."

5. References: I have not proofread the References, however, casual inspection has already picked up two errors: Ref 20 should be “Hobert” and not “Hoberst”; Ref 54 has not title.

Response: Thank you very much. We now proofread the references thoroughly.

6. Figure captions are a bit brief. For example, in Fig 3b, I have no idea what is being shown. I can see that fluorescence increases more rapidly at low pH, however, I have no idea as to the protein concentration nor any other experimental conditions.

Response: We went through the captions and provided elaborate descriptions, where needed. Thank you.

Page 36, last line: “Fig. 3. ...R18 unquenching was measured in the presence of 25 μ M CLIC5 following exposure to different pH values as indicated. (c) R18 unquenching following calcium-dependent lipid mixing by DOC2B, serving as a positive control. (d) Dithionite scrambling assay. CLIC5 does not scramble NBD-PE labeled liposomes, as reflected by the plateau at 50% prior to detergent solubilization with Triton X-100.”

7. Figure 6: in panel b, the right hand side says: “2000s (pH 5.5) -20s (pH 7.5)” on one line and “6000s (pH 5.5) -60s (pH 7.5)” on the next. Why these numbers? are the 20s and 60s significant? In contrast, panel d has “20s, 60s etc”. Why the difference in reporting? Is it significant?

We thank you for pointing this out. We admit that without further explanation, this may seem confusing. We therefore added a short explanation to the legend of Fig 6b.

*Page 38: “Fig. 3. ...**(b)** The difference in deuteration levels at the indicated time points between pH5.5 and 7.5 for CLIC5-WT (upper panel) and CLIC5-F34D (lower panel). Due to the difference in the exchange rate at different pH values (exchange 100x slower at pH 5.5 than at 7.5), time points of equal exchange (20s or 60s at pH 7.5 and 2000s or 6000s at pH 5.5, respectively) must be compared.”*

The HDX is acid-base catalyzed and thus depends on pH. Each pH shift by 1 unit means 10-fold difference in the exchange rate (PMID: 25290210). Thus, what lasts 20s at pH 7.5 can be achieved at pH 5.5 (100x slower exchange) after 2000s. Keeping that in mind, we already designed the exchange time points (20s, 60s, 300s, 2000s, 6000s) in such a manner that an exact overlap of the respective points is achieved after the pH correction. Thus, if we want to show differences between protein samples incubated at pH 5.5 and 7.5, we must stick to the matching points – 20s vs 2000s and 60s vs 6000s (300s has no match). That applies to Fig 6b. In contrast, the difference between WT and F34D shown at the same pH, either pH 5.5 or 7.5, does not require any correction and is done using the same time points. Therefore, in this case, we present the full time scale of HDX kinetics.

8. SI Fig S2: Something is not right here. Panel a looks like F34D (gold) superposed on the loop deletion mutant delta57-68. while Panel b looks like 6Y2H versus the WT deletion mutant. This is not what the caption says and it is not what the labels in the figure say.

Response: Thank you for noticing this mistake. We sincerely apologize and now made appropriate changes to the figure legend and caption.

9. SI Fig S3 panels b and c: why do you compare pH 7.5 at 60 seconds but pH 5.5 at 6,000 seconds? Is there a reason for using different time points? This seems strange.

Response: Thank you for bringing this issue to our attention. As indicated in our response to comment 7, we added the aforementioned clarification to the Figure legend of panel 6b. Briefly, due to the difference in deuterium exchange rate at different pH values, only time points that represent equal exchange can be compared.

10. SI Figure S5: these thermal shift assay plots look like the derivative of the fluorescence with respect to temperature (dF/dT) and not the normalised fluorescence as the axis labels say.

Response: Thank you for this comment. The TSA plots present the normalized fluorescence, as indicated. CLICs demonstrate high conformational flexibility in solution. Previously, we showed using small-angle X-ray scattering that both CLIC6 and CLIC5 sample a significant portion of the available conformational space, leading to the inevitable exposure of their cryptic and hydrophobic inter-domain interface. Consequently, when using SYPRO Orange as a probe, CLICs show a high initial fluorescent signal, likely due to the interaction between the dye and the exposed interface. The decline observed after reaching the fluorescence intensity peak is explained by protein phase transition and exclusion from solution due to thermal-dependent precipitation and aggregation (Niesen et al., Nature Protocols, 2007). Please note that Supplementary Figure 5 is Supplementary Figure 7 in the revised version.

Reviewer #2:

The manuscript by Manori et al. provides novel information on the role of CLIC5 as a fusogen. CLICs are generally understudied proteins, and the functional role of these proteins is not yet elucidated. These proteins exist in membrane and soluble forms and both forms have distinct roles. Authors have shown that CLIC5 interacts with the membrane and induces fusion by using several independent approaches and the data is compelling. An amino acid F34 (possibly) in the transmembrane domain was shown to ablate this fusion. Experiments in *C. elegans* are supportive of in vitro findings in the in vivo model.

Response: We thank the Reviewer for this overview of the manuscript and the useful comments and suggestions.

There are several concerns for the manuscript which should be addressed.

1. The title states, “CLIC proteins function as fusogens”. In the manuscript, the authors have only tested CLIC5 and not all the CLICs. The title does not reflect the outcome of the manuscript.
2. CLICs share homology with each other, and authors should also test whether fusogen property is exclusive for CLIC5 or is also shared by other CLIC proteins. This data will be highly beneficial for the manuscript along with DOC2B and 14-3-3.

Response: Thank you for raising these points (#1 and #2). The CLIC domain is highly conserved, and multiple structures determined to date support the high structural similarity shared by different paralogs, pointing towards functional conservation as well. To support this hypothesis, and in line with your comment, we have now added the new Supplementary Figure 2, showing that in addition to CLIC5, CLIC2, CLIC6, and EXC-4 (which we have already shown to function as a fusogen *in vivo*) also exhibit fusogenic activity *in vitro*.

3. The F34 residue is located in the putative transmembrane domain and is conserved in all the CLICs with the exception of CLIC3 (and Exl-1). CLIC3 could also serve as a control for the F34 claim.

Response: We thank the Reviewer for this suggestion. However, we do not have a readily available construct of CLIC3 in hand. Nonetheless, we have shown that in addition to CLIC5, three other CLICs function as fusogens. Importantly, differences in fusogenic activity between paralogs are hard to attribute to a single position, as can be done using site-directed mutagenesis experiments. We agree that further characterizations of both the fusogenic activity of the different paralogs as well as the dissection of functionally important positions should be pursued in future studies.

4. The joint loop (57-68) is critical for the folding and insertion of CLICs. The data is convincing, but I would recommend using CLIC5 without a putative trans-membrane domain as it will not be able to insert into the membrane. In contrast, the CLIC5 1-68 construct should promote the fusogen role.

Response: Thank you. The ‘joint loop’ is positioned within the TRX domain and our crystallographic analysis does not support its role in CLIC5 folding, at least in the globular form. Additionally, testing the fusogenic

activity of CLIC5 with deletion of this loop (CLIC5- Δ loop) revealed that it plays no significant functional role in this process (now displayed in the new Supplementary Figure 3), with fusesen activity that is similar to that of WT CLIC5. The use of an isolated putative transmembrane domain is an intriguing proposition, but it should be kept in mind that in order for membrane fusion to occur, additional protein domains are required to form a close approximation between opposing membranes. Indeed, previous attempts to generate chimeras between CLIC1 and EXC-4, swapping only the putative transmembrane domains, yielded poor membrane localization and phenotypical rescue (Berry and Hobert, 2006). This suggests that the mere insertion into the membrane is required but not sufficient to promote fusion.

5. CLIC5 is known to interact with ezrin and other cytoskeletal proteins. In *C. elegans*, did authors observe any changes in the cytoskeletal proteins that are known to interact with CLIC5? This data should be presented.

Response: The *C. elegans* ortholog of ezrin is called ERM-1. It is not known whether ERM-1 interacts with EXC-4; however, ERM-1 is known to be essential for excretory canal lumen extension (Nat Cell Biol. 2013 Feb;15(2):143-56). Therefore, to test whether the F38D mutation in EXC-4 has any effect on ERM-1 localization within the excretory canal cell, we introduced by CRISPR-Cas9 the F38D mutation in a strain (VJ402) that expresses ERM-1::GFP as an extrachromosomal array (Development. 2012 Jun;139(11):2071-83). Comparing the ERM-1::GFP localization in VJ402 with ERM-1::GFP localization in the newly created strain with the F38D mutation in EXC-4 did not reveal any difference in ERM-1 localization, ruling out the possibility that the effect of the F38D mutation is to disrupt ERM-1 localization. This new data was added to the paper in supplementary figure 10. Another cytoskeletal protein that was shown in human cells to interact with CLIC5 is plastin. We examined plastin localization in a *C. elegans* strain in which the endogenous PLST-1 was fluorescently tagged with GFP (J Cell Biol. 2017 May 1;216(5):1371-1386) and found that it is not expressed at all in the excretory canal cell and therefore did not pursue it any further.

Page 12, first paragraph: "...CLIC5 in placental microvilli was found to interact with ezrin²¹, whose worm ortholog, ERM-1, is known to be essential for excretory canal cell lumen extension⁵⁰. To test whether the F38D mutation in EXC-4 affects ERM-1 localization, we introduced the exc-4 F38D mutation in an ERM-1::GFP reporter strain⁵¹. As shown in Supplementary Figure 10, the exc-4 F38D mutation did not affect ERM-1 localization in the excretory canal, where it appeared as two parallel lines lining the lumen, similar to the control."

6. The pH data is very exciting, however, the insertion of CLICs is also redox-dependent via a cysteine residue on top of the transmembrane domain, and it will be useful to see if the cysteine is also involved in the function.

Response: Thank you for this comment. Redox-dependent regulation of CLICs was extensively studied in the past and is potentially of great functional importance. As the

Reviewer suggested, C32 seems to play a role in membrane fusion (as assessed by the carboxyfluorescein assay; see the attached Figure). We are now working on developing MS-based methodologies to decipher the mechanism of the redox-dependent effect of CLIC-mediated membrane fusion. However, as these studies will require extensive experimentation, we feel that they are beyond the scope of the current manuscript.

Minor comments

1. There are some spelling mistakes such as ‘wavelenght’

Response: Thank you. We have carefully gone over the manuscript and corrected all such typos.

2. Usually POPE and POPS are used for CLICs. Please provide an explanation of why POPG is used as PE and PG are the main lipid components of the inner bacterial membrane.

Response: To the best of our knowledge, lipid specificity has never been shown for CLICs. Moreover, the PE and PG mixture was previously used to characterize different eukaryotic proteins, such as Piezo (Guo and MacKinnon, 2017) and Slo2.2 (Hite and MacKinnon, 2017). Finally, we show *in vitro* that other proteins do not promote fusion, and show *in vivo*, and now *in vitro* as well, that EXC-4 is involved in fusion.

3. In the cartoon, protein is shown on the outer leaf of the bilayer and not spanning the membrane. Is there a specific reason for not adding protein to both membranes (spanning)?

Response: Thank you. Our results show direct protein-membrane interactions, but we have not demonstrated that the protein inserts into the membrane. Although previous studies have shown membrane insertion, we wanted to avoid over-interpretation of the data presented in this manuscript.

Reviewer #1 (Remarks to the Author):

The authors have addressed all issues raised in my review. The revised manuscript makes a major contribution to understanding the function of CLIC proteins. I recommend that this manuscript be accepted for publication in Nature Communications.